# FAST MODEL EDITING AT SCALE

**Eric Mitchell, Charles Lin, Antoine Bosselut, Chelsea Finn, Christopher D. Manning**
Stanford University
`eric.mitchell@cs.stanford.edu`

## ABSTRACT

While large pre-trained models have enabled impressive results on a variety of downstream tasks, the largest existing models still make errors, and even accurate predictions may become outdated over time. Because detecting all such failures at training time is impossible, enabling both developers and end users of such models to correct inaccurate outputs while leaving the model otherwise intact is desirable. However, the distributed, black-box nature of the representations learned by large neural networks makes producing such targeted edits difficult. If presented with only a single problematic input and new desired output, fine-tuning approaches tend to overfit; other editing algorithms are either computationally infeasible or simply ineffective when applied to very large models. To enable easy post-hoc editing at scale, we propose Model Editor Networks with Gradient Decomposition (MEND), a collection of small auxiliary editing networks that use a single desired input-output pair to make fast, local edits to a pre-trained model's behavior. MEND learns to transform the gradient obtained by standard fine-tuning, using a low-rank decomposition of the gradient to make the parameterization of this transformation tractable. MEND can be trained on a single GPU in less than a day even for 10 billion+ parameter models; once trained MEND enables rapid application of new edits to the pre-trained model. Our experiments with T5, GPT, BERT, and BART models show that MEND is the only approach to model editing that effectively edits the behavior of models with more than 10 billion parameters. Code available at https://sites.google.com/view/mend-editing.

## 1 INTRODUCTION

Increasingly large models have improved performance on a variety of modern computer vision (Huang et al., 2017; Chen et al., 2022) and especially natural language processing (Vaswani et al., 2017; Brown et al., 2020) problems. However, a key challenge in deploying and maintaining such models is issuing patches to adjust model behavior after deployment (Sinitsin et al., 2020). When a neural network produces an undesirable output, making a localized update to correct its behavior for a single input or small number of inputs is non-trivial, owing to the distributed nature of the model's representations. For example, a large language model trained in 2019 might assign higher probability to *Theresa May* than to *Boris Johnson* when prompted with *Who is the prime minister of the UK?* (see Table 2 for an example with a real large language model; see Lazaridou et al. (2021) for a systematic study of failures of temporal generalization in LMs). An ideal model editing

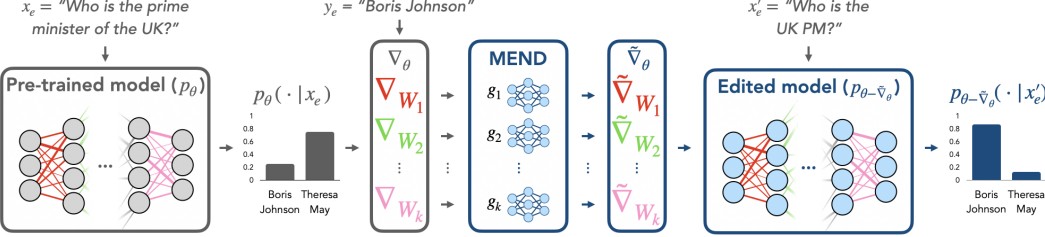

**Figure 1:** The proposed algorithm MEND enables editability by training a collection of MLPs to modify model gradients to produce *local* model edits that do not damage model performance on unrelated inputs. MEND is efficient to train and apply edits, even for very large models, as shown in Section 5.1.

procedure could quickly update the model parameters to increase the relative likelihood of *Boris Johnson* without changing the model output for unrelated inputs. This procedure would produce edits with *reliability*, successfully changing the model's output on the problematic input (e.g., *Who is the prime minister of the UK?*); *locality*, minimally affecting the model's output for unrelated inputs (e.g., *What sports team does Messi play for?*); and *generality*, generating the correct output for inputs related to the edit input (e.g., *Who is the UK PM?*).

A simple approach to making such edits is additional fine-tuning with a new label on the single example to be corrected. Yet fine-tuning on a single example tends to overfit, even when constraining the distance between the pre- and post-fine-tuning parameters (Zhu et al., 2020; De Cao et al., 2021). This overfitting leads to failures of both locality and generality. While fine-tuning on the edit example along with continued training on the training set better enforces locality, our experiments show that it still lacks generality. Further, it requires persistent access to the full training set during test time and is more computationally demanding. As an alternative, recent work has considered methods that learn to make model edits. Sinitsin et al. (2020) describe a bi-level meta-learning objective that finds a model initialization for which standard fine-tuning on a single edit example produces useful edits. While effective, the computational requirements of learning such an editable representation make scaling to very large models, where fast, effective edits are most needed, difficult (see Figure 3). De Cao et al. (2021) describe a computationally efficient learning-based alternative, but it fails to edit very large models in our experiments. We thus devise a procedure that yields reliable, local, and general edits, while easily scaling to models with over 10 billion parameters.

Our approach trains lightweight *model editor networks* to produce edits to a pre-trained model's weights when provided with the standard fine-tuning gradient of a given correction as input, leveraging the gradient as an information-rich starting point for editing (see Figure 1). Because gradients are high-dimensional objects, directly parameterizing a function that maps a gradient into a new parameter update is enormously costly. Even for a single $d \times d$ weight matrix, a naive implementation requires a mapping from $\mathbb{R}^{\mathcal{O}(d^2)} \to \mathbb{R}^{\mathcal{O}(d^2)}$, which is impractical for large models where $d \approx 10^4$. However, by *decomposing* this gradient into its rank-1 outer product form, our approach is instead able to learn a function $g : \mathbb{R}^{\mathcal{O}(d)} \to \mathbb{R}^{\mathcal{O}(d)}$. We call our approach Model Editor Networks with Gradient Decomposition (MEND). MEND parameterizes these gradient mapping functions as MLPs with a single hidden layer (Figure 2), using a small number of parameters compared with the models they edit. MEND can be applied to any pre-trained model, regardless of pre-training.

The primary contribution of this work is a scalable algorithm for fast model editing that can edit very large pre-trained language models by leveraging the low-rank structure of fine-tuning gradients. We perform empirical evaluations on a variety of language-related tasks and transformer models, showing that MEND is the only algorithm that can consistently edit the largest GPT-style (Radford et al., 2019; Black et al., 2021; Wang and Komatsuzaki, 2021) and T5 (Raffel et al., 2020) language models. Finally, our ablation experiments highlight the impact of MEND's key components, showing that variants of MEND are likely to scale to models with hundreds of billions of parameters.

## 2 THE MODEL EDITING PROBLEM

The goal of model editing is to enable the use of a single pair of input $x_e$ and desired output $y_e$ to alter a *base model*'s output for $x_e$ as well as its *equivalence neighborhood* (related input/output pairs), all while leaving model behavior on unrelated inputs unchanged (Sinitsin et al., 2020; De Cao et al., 2021). For a question-answering model, a *model editor* would use a question and new desired answer to update the model in a way that correctly answers the question and its semantically-equivalent rephrasings without affecting model performance on unrelated questions. Some model editors, including ours, use a training phase before they can apply edits (Sinitsin et al., 2020; De Cao et al., 2021), using an edit training dataset $D_{edit}^{tr}$ that specifies the types of edits that will be made.

More precisely, the base model $f_\theta : \mathcal{X} \times \Theta \to \mathcal{Y}$ is a differentiable function that maps an input $x$ and set of parameters $\theta$ to an output $y$. A model editor is a function $E : \mathcal{X} \times \mathcal{Y} \times \mathcal{L} \times \Theta \times \Phi \to \Theta$ that maps an *edit input* $x_e$, *edit label* $y_e$ (a class label or sequence of tokens), loss function $l_e : \mathcal{X} \times \mathcal{Y} \times \Theta \to \mathbb{R}$, base model parameters $\theta$, and optional editor parameters $\phi$ to a new set of model parameters $\theta_e$. We use the loss function $l_e(x, y, \theta) = -\log p_\theta(y|x)$, based on past work (De Cao et al., 2021), but other choices are possible. Model editors are evaluated on a held-out dataset $D_{edit}^{te} = \{(x_e, y_e, x_{loc}, x_e', y_e')_i\}$. For algorithms that learn model editor parameters $\phi$, a dataset $D_{edit}^{tr}$ containing tuples similar to $D_{edit}^{te}$ is used, typically much smaller than the pre-trained

## MEND Architecture

**Figure 2:** The MEND architecture, consisting of two consecutive blocks, both initialized to compute the exact identity function. **Left.** The input to a MEND network is $\{\delta_{\ell+1}, u_\ell\}$, the components of the rank-1 gradient. **Right.** A MEND network produces a new rank-1 update $\tilde{\nabla}_{W_\ell}$, which is added to weights $W_\ell$ to edit the model.

model's original training set. The locality input $x_{\text{loc}}$ is simply a randomly sampled input that is used to quantify the extent to which model predictions change for unrelated inputs. The alternative edit input and label $x'_{\text{e}}$ and $y'_{\text{e}}$ are sampled from the *equivalence neighborhood* $N(x_{\text{e}}, y_{\text{e}})$ of $x_{\text{e}}$ and $y_{\text{e}}$, the set of examples that the edited model should generalize to after performing an edit with $x_{\text{e}}, y_{\text{e}}$. For $x_{\text{e}}, y_{\text{e}} = $ *Who is the prime minister of the UK? Boris Johnson*, $N(x_{\text{e}}, y_{\text{e}})$ might contain $x'_{\text{e}}, y'_{\text{e}} = $ *Who is the UK PM? Boris Johnson*, among others. $x_{\text{loc}}$ might be *What team does Messi play for?*.

In this work, we call a model editor *reliable* if the post-edit model predicts the edit label $y_{\text{e}}$ for the edit input $x_{\text{e}}$. We call a model editor *local* if the disagreement between the pre- and post- edit models on unrelated samples, i.e., $\mathbb{E}_{x_{\text{loc}} \sim D^{te}_{edit}} \text{KL}(p_\theta(\cdot|x_{\text{loc}}) \| p_{\theta_e}(\cdot|x_{\text{loc}}))$, is small.[1] Finally, we say a model editor *generalizes* if the post-edit model predicts the label $y'_{\text{e}}$ when conditioned on $x'_{\text{e}}$, for $(x'_{\text{e}}, y'_{\text{e}}) \in N(x_{\text{e}}, y_{\text{e}})$. We call a model editor *efficient* if the time and memory requirements for computing $\phi$ and evaluating $E$ are small. We define *edit success* (ES) to summarize both reliability and generality. It is measured as the average accuracy of the edited model $p_{\theta_e}$ on the edit input as well as inputs drawn uniformly from the equivalence neighborhood:

$$\text{ES} = \mathbb{E}_{x'_{\text{e}}, y'_{\text{e}} \sim N(x_{\text{e}}, y_{\text{e}}) \cup \{(x_{\text{e}}, y_{\text{e}})\}} \mathbb{1}\{\text{argmax}_y\, p_{\theta_e}(y|x'_{\text{e}}) = y'_{\text{e}}\}. \tag{1}$$

## 3 MODEL EDITOR NETWORKS WITH GRADIENT DECOMPOSITION

Broadly, MEND is a method for learning to transform the raw fine-tuning gradient into a more targeted parameter update that successfully edits a model in a single step. MEND uses $f_\theta$ and an edit training set $D^{tr}_{edit}$ to produce a collection of model editor networks $g_\ell$, which edit the model's weights given new edit pairs $(x_{\text{e}}, y_{\text{e}})$ at test time. Each $g_\ell$ transforms the fine-tuning gradient for a particular layer $\ell$ into a parameter update for the layer that provides the reliability, locality, generality, and efficiency properties described earlier. Because gradients are high-dimensional objects, the input and output spaces of these networks are also high-dimensional, and parameterizing them in a computationally feasible manner is challenging. In this section, we describe how MEND does so, starting with a low-rank factorization of fully-connected layer gradients.

### 3.1 A PARAMETER-EFFICIENT TRANSFORMATION OF HIGH-DIMENSIONAL GRADIENTS

The input to a MEND network $g_\ell$ is the fine-tuning gradient $\nabla_{W_\ell} l_e(x_{\text{e}}, y_{\text{e}}, \theta)$ at layer $\ell$ and the output is the layer's parameter edit, which we call $\tilde{\nabla}_{W_\ell}$. As noted earlier, for a $d \times d$ weight matrix, this function has $d^2$ inputs and outputs. Even if $g_\ell$ is a linear network with no hidden layers and produces only a rank-1 parameter edit (motivated by the effectiveness of low-rank model edits observed by Hu et al. (2021)), this function would still require $d^2(d + d) = 2d^3$ parameters. For a low-rank linear parameterization of $g_\ell$ with rank $r$, we have $r(d^2 + 2d)$ parameters, which still carries an unacceptable cost for non-trivial $r$, considering that $d \approx 10^4$ for some models (Raffel et al., 2020).

MEND solves this problem using the fact that the input to $g_\ell$, the fine-tuning gradient, is a rank-1 matrix: the gradient of loss $L$ with respect to weights $W_\ell$ in layer $\ell$ of an MLP is a rank-1 matrix for each of $B$ batch elements $\nabla_{W_\ell} L = \sum_{i=1}^B \delta^i_{\ell+1} u^{i\top}_\ell$, where $\delta^i_{\ell+1}$ is the gradient of the loss for batch element $i$ with respect to the preactivations at layer $\ell + 1$, and $u^i_\ell$ are the inputs to layer $\ell$

---

[1] See Appendix C.2 for additional details on estimating this KL-divergence.

| **Algorithm 1** MEND Training | **Algorithm 2** MEND Edit Procedure |
|---|---|
| 1: **Input:**  Pre-trained $p_{\theta_W}$, weights to make editable $\mathcal{W}$, editor params $\phi_0$, edit dataset $D_{edit}^{tr}$, edit-locality tradeoff $c_{\text{edit}}$
2: **for** $t \in 1, 2, ...$ **do**
3:  Sample $x_{\text{e}}, y_{\text{e}}, x'_{\text{e}}, y'_{\text{e}}, x_{\text{loc}} \sim D_{edit}^{tr}$
4:  $\tilde{\mathcal{W}} \leftarrow \text{EDIT}(\theta_{\mathcal{W}}, \mathcal{W}, \phi_{t-1}, x_{\text{e}}, y_{\text{e}})$
5:  $L_{\text{e}} \leftarrow -\log p_{\theta_{\tilde{\mathcal{W}}}}(y'_{\text{e}}\vert x'_{\text{e}})$
6:  $L_{\text{loc}} \leftarrow \text{KL}(p_{\theta_{\mathcal{W}}}(\cdot\vert x_{\text{loc}})\Vert p_{\theta_{\tilde{\mathcal{W}}}}(\cdot\vert x_{\text{loc}}))$
7:  $L(\phi_{t-1}) \leftarrow c_{\text{edit}}L_{\text{e}} + L_{\text{loc}}$
8:  $\phi_t \leftarrow \text{Adam}\left(\phi_{t-1}, \nabla_\phi L(\phi_{t-1})\right)$ | 1: **procedure** $\text{EDIT}(\theta, \mathcal{W}, \phi, x_{\text{e}}, y_{\text{e}})$
2:  $\hat{p} \leftarrow p_{\theta_{\mathcal{W}}}(y_{\text{e}}\vert x_{\text{e}})$, **caching** input $u_\ell$ to $W_\ell \in \mathcal{W}$
3:  $L(\theta, \mathcal{W}) \leftarrow -\log \hat{p}$   ▷ Compute NLL
4:  **for** $W_\ell \in \mathcal{W}$ **do**
5:   $\delta_{\ell+1} \leftarrow \nabla_{W_\ell u_\ell + b_\ell} l_{\text{e}}(x_{\text{e}}, y_{\text{e}})$ ▷ Grad wrt output
6:   $\tilde{u}_\ell, \tilde{\delta}_{\ell+1} \leftarrow g_{\phi_\ell}(u_\ell, \delta_{\ell+1})$ ▷ Pseudo-acts/deltas
7:   $\tilde{W}_\ell \leftarrow W_\ell - \tilde{\delta}_{\ell+1}\tilde{u}_\ell^\top$   ▷ Layer $\ell$ model edit
8:  $\tilde{\mathcal{W}} \leftarrow \{\tilde{W}_1, ..., \tilde{W}_k\}$
9:  **return** $\tilde{\mathcal{W}}$   ▷ Return edited weights |

for batch element $i$ (see Appendix D). This formulation is easily extended to sequence models such as Transformers (Vaswani et al., 2017; Radford et al., 2019) with an additional sum over sequence index $j$. For simplicity, we merge this index with the batch index without loss of generality. This decomposition enables a network to condition directly on the gradient of a single example with only $2d$ (rather than $d^2$) input neurons.[2] With this parameterization, MEND learns functions $g_\ell$, with parameters $\phi_\ell$, which map $u_\ell^i$ and $\delta_{\ell+1}^i$ to *pseudoactivations* $\tilde{u}_\ell^i$ and *pseudodelta* $\tilde{\delta}_{\ell+1}^i$. The model edit for weight matrix $W_\ell$ is then

$$\tilde{\nabla}_{W_\ell} = \sum_{i=1}^B \tilde{\delta}_{\ell+1}^i \tilde{u}_\ell^{i\top}. \tag{2}$$

To further reduce the number of additional parameters, MEND shares parameters across editor networks $g_\ell$ (note Figure 2 omits this for clarity). Because the sizes of $u_\ell$ and $\delta_{\ell+1}$ depend on the shape of the weight matrix $W_\ell$, MEND learns a separate set of editor parameters for each unique *shape* of weight matrix to be edited. Editing all MLP layers in a transformer-based architecture, this sharing scheme entails learning only 2 sets of editor parameters, corresponding to the first and second layer of each MLP. To enable some layer-wise specialization, MEND applies a layer-specific scale $s_\ell$ and offset $o_\ell$ to the editor network hidden state and output, similar to FiLM layers (Perez et al., 2018). Putting everything together, a MEND network computes $g_\ell(z_\ell)$ where $z_\ell = \text{concat}(u_\ell, \delta_{\ell+1})$ as

$$h_\ell = z_\ell + \sigma(s_\ell^1 \odot (U_1 V_1 z_\ell + b) + o_\ell^1), \qquad g(z_\ell) = h_\ell + \sigma(s_\ell^2 \odot U_2 V_2 h_\ell + o_\ell^2) \tag{3a,b}$$

where $\sigma$ is a non-linear activation function s.t. $\sigma(0) = 0$ (ReLU in this work) and $U_j, V_j$ correspond to a low rank factorization of MEND's weights at layer $j$ (keeping MEND's total parameters $O(d)$).

To summarize, MEND parameterizes $g_\ell$ as an MLP with low-rank weight matrices, residual connections, and a single hidden layer (see Figure 2). To edit layer $\ell$, layer activations $u_\ell^i$ and output gradients $\delta_{\ell+1}^i$ are concatenated and passed together to $g_\ell$, producing a vector of equal size, which is split into pseudoactivations $\tilde{u}_\ell^i$ and pseudodeltas $\tilde{\delta}_{\ell+1}^i$, ultimately producing $\tilde{\nabla}_{W_\ell}$ (Eq. 2). The final edited weights are $\tilde{W} = W_\ell - \alpha \tilde{\nabla}_{W_\ell}$, where $\alpha_\ell$ is a learned per-layer (scalar) step size.

### 3.2 Training MEND

MEND uses an editing training set $D_{edit}^{tr}$ to learn parameters $\phi_\ell$ for each of the MEND networks $g_\ell$. Before training, we select the weights of the model $\mathcal{W} = \{W_1, ..., W_M\}$ that we would like to make editable (e.g., the weight matrices in the last $M$ layers). At each step of training, we sample an edit example $(x_{\text{e}}, y_{\text{e}})$, locality example $x_{\text{loc}}$, and equivalence examples $(x'_{\text{e}}, y'_{\text{e}})$ from the edit train set $D_{edit}^{tr}$. Recall that $x_{\text{loc}}$ is sampled independently from the edit example, so that it is very likely that it is unrelated to the edit example. We use $(x_{\text{e}}, y_{\text{e}})$ to compute the raw gradient $\nabla_{W_\ell} p_{\theta_{\mathcal{W}}}(y_{\text{e}}\vert x_{\text{e}})$ for each weight matrix $W_\ell \in \mathcal{W}$, using $\theta_{\mathcal{W}}$ to denote the model parameters with un-edited weights. We then compute the parameter update for each layer $\tilde{W} = W_\ell - \alpha_\ell \tilde{\nabla}_{W_\ell}$ ($\tilde{\nabla}_{W_\ell}$ from Eq. 2).

We compute the training losses for MEND using the edited model parameters $\tilde{\mathcal{W}}$, which we back-propagate into the editing networks. Note that we do not compute any higher-order gradients, because we do not optimize the pre-edit model parameters. The training losses are $L_{\text{e}}$, which measures edit success and $L_{\text{loc}}$, which measures edit locality (the KL divergence between the pre-edit and post-edit model conditioned on the locality input $x_{\text{loc}}$), defined as follows (also Alg. 1 lines 5–7):

---

[2]For a batch/sequence, we transform the gradient for each batch/sequence element independently and sum the result to acquire the final transformed gradient for the entire batch/sequence.

> **MEND losses:** $\quad L_{\mathrm{e}} = -\log p_{\theta_{\widetilde{\mathcal{W}}}}(y_{\mathrm{e}}'|x_{\mathrm{e}}'), \quad L_{\mathrm{loc}} = \mathrm{KL}(p_{\theta_{\mathcal{W}}}(\cdot|x_{\mathrm{loc}})\|p_{\theta_{\widetilde{\mathcal{W}}}}(\cdot|x_{\mathrm{loc}})). \quad$ (4a,b)

Intuitively, $L_{\mathrm{e}}$ is small if the model has successfully updated its output for the edit example's equivalence neighborhood, while $L_{\mathrm{loc}}$ is small if the edit did not affect the model's behavior on unrelated inputs. The total training loss for a MEND network is computed as $L_{\mathrm{MEND}} = c_{\mathrm{e}}L_{\mathrm{e}}(\theta_{\widetilde{\mathcal{W}}}) + L_{\mathrm{loc}}(\theta_{\mathcal{W}}, \theta_{\widetilde{\mathcal{W}}})$. We optimize $L_{\mathrm{MEND}}$ with respect to the MEND parameters at each time step using the Adam optimizer (Kingma and Ba, 2015), using $c_{\mathrm{e}} = 0.1$ for all experiments.

While MEND's parameterization can tractably *represent* a mapping from gradients to model edits, training the editor presents its own challenges. Appendix A describes MEND's identity initialization and input normalization, which our ablations in Section 5.4 show are important to effective edits.

## 4 RELATED WORK

Various strategies for model editing exist, including modifications of standard fine-tuning intended to enforce locality by reducing distance traveled in parameter space (Zhu et al., 2020) or even find the min-L2 norm parameter update that reliably edits the model's output (Sotoudeh and Thakur, 2021). However, De Cao et al. (2021) observe that parameter-space constraints do not always translate to useful function-space constraints for neural networks. Our fine-tuning baselines thus use a KL-divergence constraint in function space, but, even with this modification, we find that fine-tuning generally doesn't consistently provide edit generality. Other approaches to editing such as Editable Neural Networks (**ENN**; Sinitsin et al. (2020)) or KnowledgeEditor (**KE**; De Cao et al. (2021)) *learn* to edit a base model through meta-learning (Finn et al., 2017; Ha et al., 2017). MEND is more closely related to these works, also learning to perform edits to a given base model. MEND dif-

| Editor | Preserves model? | Only $(x_{\mathrm{e}}, y_{\mathrm{e}})$? | Batched edits? | Scales to 10B? | Few steps? |
|---|---|---|---|---|---|
| FT | ✓ | ✓ | ✓ | ✓ | ✗ |
| FT+KL | ✓ | ✗ | ✓ | ✓ | ✗ |
| ENN | ✗ | ✓ | ✓ | ✗ | ✓ |
| KE | ✓ | ✓ | ? | ✓ | ✓ |
| MEND | ✓ | ✓ | ✓ | ✓ | ✓ |

Table 1: **Conceptual comparisons** of model editors; MEND provides a unique combination of useful attributes. **Preserves model** means the editor guarantees model predictions will not be altered *before* an edit is applied. **Only** $(x_{\mathbf{e}}, y_{\mathbf{e}})$ means the editor applies an edit at test time using only the edit pair (not needing access to the training set at test time as well). **Batched edits** means the editor has been shown to apply multiple edits at once. **Scales to 10B** means our implementation of the editor could run on a model with over 10B parameters using our single-GPU environment (see Appendix C.3). **Few steps** means edits are applied with one or a small number of steps. **FT** refers to fine-tuning; **FT+KL** adds a KL-divergence penalty between the original and fine-tuned model.

fers from ENN as it does not further train (and thus modify) the base model before an edit is needed, and it does not compute higher-order gradients. Because ENN modifies the pre-edit model, the training process retains a copy of the original model in order to enforce the constraint that the editable model agrees with the original pre-trained model's predictions. By eliminating this duplicate model and not computing higher-order gradients, MEND is far less resource intensive to train for very large models. Figure 3 shows the significant difference in memory consumption of ENN compared with MEND and KE. MEND is most similar to KE, which also presents a first-order algorithm that does not modify the pre-edit model. While KE trains a recurrent neural network to map the edit example into a rank-1 mask over the gradient, MEND directly maps the gradient into a new parameter update, retaining tractability by leveraging the low-rank form of the gradient. Table 1 contains an overview of algorithmic tradeoffs. See Appendix B for extended discussion of related work.

Various methods for meta-learning also use gradient transforms to achieve better model updates for few-shot learning (Ravi and Larochelle, 2017; Li et al., 2017; Lee and Choi, 2018; Park and Oliva, 2019; Flennerhag et al., 2020). However, these approaches do not leverage the factorized gradient, limiting them to simpler transformations (typically linear) of the gradient and/or transformations that also often impact the function computed by the forward pass of the model. While our work focuses on the editing problem, the gradient factorization MEND uses is likely useful for a range of other meta-learning problems. Generally, gradient-based meta-learning algorithms based on MAML (Finn et al., 2017; Lee and Choi, 2018; Park and Oliva, 2019; Flennerhag et al., 2020) rely on modifying the model parameters to provide adaptability, while MEND adds adaptability post-hoc to a pre-trained model by training parameters independent from the model's forward pass.

In the NLP literature, many papers have investigated the locus of various types of knowledge in language models, using learned probe models or iterative search procedures to test for linguistic structures (Belinkov et al., 2017; Conneau et al., 2018; Hewitt and Manning, 2019) or facts about

| Input | Pre-Edit Output | Edit Target | Post-Edit Output |
|---|---|---|---|
| 1a: **Who is India's PM?** | Satya Pal Malik ✗ | **Narendra Modi** | Narendra Modi ✓ |
| 1b: **Who is the prime minister of the UK?** | Theresa May ✗ | **Boris Johnson** | Boris Johnson ✓ |
| 1c: Who is the prime minister of India? | Narendra Modi ✓ | — | Narendra Modi ✓ |
| 1d: Who is the UK PM? | Theresa May ✗ | — | Boris Johnson ✓ |
| 2a: **What is Messi's club team?** | Barcelona B ✗ | **PSG** | PSG ✓ |
| 2b: **What basketball team does Lebron play on?** | Dallas Mavericks ✗ | **the LA Lakers** | the LA Lakers ✓ |
| 2c: Where in the US is Raleigh? | a state in the South ✓ | — | a state in the South ✓ |
| 3a: **Who is the president of Mexico?** | Enrique Pea Nieto ✗ | **Andrés Manuel López Obrador** | Andrés Manuel López Obrador ✓ |
| 3b: Who is the vice president of Mexico? | Yadier Benjamin Ramos ✗ | — | Andrés Manuel López Obrador ✗ |

**Table 2: Examples of using MEND** to edit a T5-small model fine-tuned on Natural Questions by Roberts et al. (2020). Each example shows the output of the model before and after editing. **Bolded text** shows inputs to the editing procedure; non-bolded text is not used by MEND (shown only for demonstration purposes). In examples 1 and 2, we perform multiple edits in sequence with MEND; in ex. 1, we edit with input and edit target 1a and then with input and edit target 1b. Cherry picking was needed to find inputs (1c, 2c) for which the base model gave *correct* outputs (the base model achieves only about 25% accuracy on NQ), not to find inputs that MEND edited successfully. See Table 10 in the Appendix for additional examples and failure cases.

the world (Petroni et al., 2019; Jiang et al., 2020; Dai et al., 2021). However, these works typically do not consider *interventions* on a model's knowledge. Exceptions are Dai et al. (2021) and Wang et al. (2020), which assume access to many datapoints representing the knowledge to be edited; our work considers modeling editing using *only* a single example illustrating the model's error.

## 5 EXPERIMENTS

A key motivation for MEND is scalability to large models, which requires an algorithm to be efficient in terms of computation time and particularly memory consumption. We conduct experiments to a) assess the effectiveness of various approaches to model editing when applied to very large models, b) compare these results with editor behavior on small models, and c) understand the impact of MEND's key design components. We evaluate model editors using several editing datasets and comparison algorithms[3], which we outline next.

**Editing Datasets.** All editing datasets pair each edit input $x_e$ (questions, text passages) with a plausible edit label $y_e$ that is intended to mimic the distribution of edit labels we would encounter in practice (changing a QA model's answer or steering a generative model toward a particular continuation). For example, in a QA setting, plausible edit labels include the ground truth label as well as entities of the same type as the true answer. See Appendix C.4 Tables 7 and 8 for sample data. Specifically, for seq2seq models, we use the **zsRE question-answering** dataset (Levy et al., 2017) using question rephrasings generated by backtranslation as the equivalence neighborhood and train/val splits generated by De Cao et al. (2021). Each $x_e$ is a question about an entity, and plausible alternative edit labels $y_e$ are sampled from the top-ranked predictions of a BART-base model trained on zsRE question-answering. When editing models pre-trained on the zsRE question-answering problem, we sample $x_{loc}$ as independent questions from the edit train set. For other experiments (Section 5.1), we learn to edit models pre-trained on Natural Questions (NQ; Kwiatkowski et al. (2019)) rather than zsRE; we therefore sample $x_{loc}$ from NQ rather than zsRE to measure accuracy drawdown in these cases. For classification models (e.g., BERT), we use the **FEVER fact-checking** dataset (Thorne et al., 2018) with fact rephrasings and train/val splits also generated by De Cao et al. (2021). Each $x_e$ is a fact, and each $y_e$ is a random binary label sampled from a Bernoulli distribution with $p = 0.5$. Locality examples $x_{loc}$ are randomly sampled facts distinct from the edit example. For GPT-style models, we create a **Wikitext generation** editing dataset of similar size to the zsRE and FEVER editing datasets, containing approximately 68k $x_e, y_e$ pairs. Each $x_e$ is a passage sampled

---

[3]For each dataset, **all algorithms edit the same parameters**. For BART/T5, we edit the MLP layers of the last 2 encoder & decoder blocks; for GPT/BERT models, we edit the MLPs in the last 3 blocks.

| | Wikitext Generation | | | | zsRE Question-Answering | | | |
| --- | --- | --- | --- | --- | --- | --- | --- | --- |
| | GPT-Neo (2.7B) | | GPT-J (6B) | | T5-XL (2.8B) | | T5-XXL (11B) | |
| Editor | ES ↑ | ppl. DD ↓ | ES ↑ | ppl. DD ↓ | ES ↑ | acc. DD ↓ | ES ↑ | acc. DD ↓ |
| FT | 0.55 | 0.195 | 0.80 | 0.125 | 0.58 | **< 0.001** | 0.87 | **< 0.001** |
| FT+KL | 0.40 | **0.026** | 0.36 | 0.109 | 0.55 | **< 0.001** | 0.85 | **< 0.001** |
| KE | 0.00 | 0.137 | 0.01 | 0.068 | 0.03 | **< 0.001** | 0.04 | **< 0.001** |
| MEND | **0.81** | 0.057 | **0.88** | **0.031** | **0.88** | 0.001 | **0.89** | **< 0.001** |

**Table 3: Editing very large pre-trained models** on our Wikitext generative editing problem and the zsRE question-answering editing problem used by De Cao et al. (2021). MEND consistently produces more effective edits (higher success, lower drawdown) than existing editors. **ES** is the edit success rate, while **ppl. DD** and **acc. DD** are the model drawdown in units of perplexity increase or accuracy decrease, respectively. Due to ENN's memory requirements, we were unable to run the algorithm for models of this size. The low drawdown for all T5 models may occur because the T5 models (pre-trained on mask filling and finetuned for question-answering by Roberts et al. (2020)) might not be fully converged on the question-answering problem. Edits may therefore effectively serve as task specification, further fine-tuning the model on question-answering. **FT** refers to fine-tuning; **FT+KL** is fine-tuning with a KL-div. penalty between the original and fine-tuned model.

from Wikitext-103 and $y_e$ is a 10-token sample from a pre-trained distilGPT-2 model.[4] $x_{loc}$ is chosen depending on the pre-trained model: for models pre-trained on Wikitext, $x_{loc}$ is sampled from Wikitext-103 (independently from $x_e$). For GPT-Neo/J, we sample $x_{loc}$ from OpenWebText (OWT; (Gokaslan and Cohen, 2019)) to better match the model's original training data. The equivalence neighborhood in this setting is $N(x_e, y_e) = \{(x_e^k, y_e)\}$, where $x_e^k$ is formed by removing a prefix of up to $\frac{|x_e|}{2}$ tokens from the beginning of $x_e$, where $|x_e|$ is the length of $x_e$ in tokens.

**Comparison of model editors.** We compare MEND with several other model editors, including two fine-tuning-based algorithms (which do not train any model editor at all) and two learned model editors. The **fine-tune (FT)** algorithm fine-tunes on the edit example $(x_e, y_e)$ until the label is assigned the highest likelihood (using greedy decoding for sequence models). The 'oracle' **fine-tune + KL (FT+KL)** algorithm has access to the training set at test time and adds $L_{loc}$ (Eq. 4b) to the test-time fine-tuning objective (which is typically only computable during model editor training). Similarly to De Cao et al. (2021), we limit each of these algorithms to 100 fine-tuning steps. Additionally, we compare with two *learned* model editors: a re-implementation of Editable Neural Networks (ENN; Sinitsin et al., 2020) when possible (due to high memory usage) and KnowledgeEditor (KE; De Cao et al., 2021). We use **identical hyperparameters** for MEND across all models and datasets. For BART and T5 models, we edit the MLP weight matrices in the last 2 transformer blocks of the encoder and decoder; for other models, we edit the MLP weights in the last 3 transformer blocks. Appendix G explores a simple caching-based model editor that stores model edits in memory.

**Metrics.** Our experiments measure the reliability and generality of a model editor using **edit success (ES)** (Eq. 1). To assess locality, we use **drawdown (DD)**, which is defined as the performance degradation of the edited model on the rest of the dataset, measured as either the edited model's perplexity increase or accuracy decrease compared to the base model, depending on the problem.

## 5.1 Editing very large transformer models

We first consider the problem of editing some of the largest publicly-available Transformer models. We use GPT-Neo (2.7B parameters; Black et al., 2021) and GPT-J (6B parameters; Wang and Komatsuzaki, 2021), several times larger than GPT-2 (Radford et al., 2019), and the largest two T5 models, T5-XL (2.8B parameters) and T5-XXL (11B parameters) fine-tuned on NQ (Roberts et al., 2020). Table 3 shows the results; MEND provides the most successful edits across tasks. Fine-tuning achieves lower edit success on the Wikitext task and exhibits a much larger perplexity increase than MEND. On the question-answering edit task, fine-tuning shows similarly reduced edit success, struggling to generalize to some rephrasings of the edit input. The KL-constrained baseline reduces the perplexity drawdown for GPT-Neo and GPT-J, but at the cost of edit success. KE is ineffective at this scale, generally failing to provide successful edits. For these experiments, we use OWT and NQ to measure drawdown for generation and question-answering, respectively, as they are more representative of the data used to train the base models.

---

[4]The base model's greedy 10-token prediction agrees with these edit targets for <1% of examples.

| | FEVER Fact-Checking | | zsRE Question-Answering | | Wikitext Generation | |
|---|---|---|---|---|---|---|
| | BERT-base (110M) | | BART-base (139M) | | distilGPT-2 (82M) | |
| Editor | ES ↑ | acc. DD ↓ | ES ↑ | acc. DD ↓ | ES ↑ | ppl. DD ↓ |
| FT | 0.76 | **< 0.001** | 0.96 | **< 0.001** | 0.29 | 0.938 |
| FT+KL | 0.64 | **< 0.001** | 0.89 | **< 0.001** | 0.17 | **0.059** |
| ENN | **0.99** | 0.003 | **0.99** | **< 0.001** | **0.93** | 0.094 |
| KE | 0.95 | 0.004 | **0.98** | **< 0.001** | 0.25 | 0.595 |
| MEND | **> 0.99** | **< 0.001** | **0.98** | 0.002 | 0.86 | 0.225 |

**Table 4: Small-scale model editing** with various model editors on three editing problems. ENN and MEND show the most consistently good performance, with ENN exceeding MEND's performance on the Wikitext problem. MEND's primary advantages are its consistent performance from 100M to 10B parameter models and the fact that it does not modify the pre-edit model (unlike ENN). The pre-trained models and editing data for the FEVER fact-checking and zsRE question-answering problems are used from the checkpoints and data released by De Cao et al. (2021); for generation, we use distilGPT-2 fine-tuned on Wikitext2 (Ma, 2021).

## 5.2 SMALLER SCALE EDITING

We conduct an additional experiment editing the BERT-base and BART-base models fine-tuned by De Cao et al. (2021) on the FEVER fact-checking and zsRE question-answering tasks, respectively, and our Wikitext editing task, editing a smaller distilGPT-2 model (Wolf et al., 2019) fine-tuned on Wikitext2 (Ma, 2021). These models are 1–2 orders of magnitude smaller than those in Section 5.1. Results are presented in Table 4. At small scale where computational requirements are not a concern, ENN is competitive with MEND, providing the best performance on the Wikitext problem. Fine-tuning overfits even more severely than with larger models, showing lower edit success (overfitting to the edit example) and higher drawdown (degrading the model more seriously). One difficulty of using ENN is that the pre-trained model itself must be fine-tuned to 'provide' editability, potentially changing the model's predictions even *before* an edit has been applied. Un-

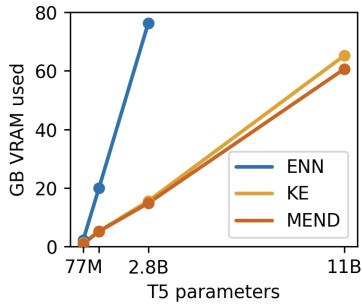

**Figure 3: GPU VRAM consumption** during training. ENN's memory usage[5] is prohibitively high for very large models, while MEND and KE can be trained on a single GPU. Figure 4 shows similar chart for GPT models.

like the large-scale experiments, drawdown is computed using samples from the same datasets as edit inputs, again in order to best match the data distribution the base models were fine-tuned on. See Appendix G for additional comparisons with the caching-based editor, which shows strong performance for zsRE and FEVER, but generally fails for Wikitext, as well as a more difficult version of the zsRE problem for which MEND still produces meaningful edits.

## 5.3 BATCHED EDITING

Table 5 compares MEND with ENN (the strongest comparison method) in a more realistic setting when multiple simultaneous zsRE QA model edits are needed; MEND consistently provides significantly more effective edits in the multi-edit setting. Both algorithms are trained and evaluated on applying $k$ simultaneous edits, with $k \in \{1, 5, 25, 75, 125\}$. MEND applies simultaneous edits by simply summing the parameter edit computed separately for each edit example. MEND applies 25 edits in a single model update with 96% edit success and less than 1% accuracy degrada-

| | Edit Success ↑ | | Acc. Drawdown ↓ | |
|---|---|---|---|---|
| Edits | ENN | MEND | ENN | MEND |
| 1 | 0.99 | 0.98 | < 0.001 | 0.002 |
| 5 | 0.94 | 0.97 | 0.007 | 0.005 |
| 25 | 0.35 | 0.89 | 0.005 | 0.011 |
| 75 | 0.16 | 0.78 | 0.005 | 0.011 |
| 125 | 0.11 | 0.67 | 0.006 | 0.012 |

**Table 5: Batched edits with MEND and ENN** on zsRE QA using the BART-base pre-trained model from De Cao et al. (2021). When applying multiple edits at once, MEND is far more effective than ENN.

tion (35% edit success for ENN), and successfully applies 67% of edits when applying 125 edits at once (11% success for ENN, although ENN's accuracy drawdown is slightly lower).

---

[5]We report the memory usage of our re-implementation of ENN (Sinitsin et al., 2020). Techniques like gradient checkpointing can reduce memory consumption, but an optimized ENN implementation is not available.

| MEND Variant | Editor Parameters | Wikitext Generation | | zsRE Question-Answering | |
|---|---|---|---|---|---|
| | | distilGPT-2 (82M) | | BART-base (139M) | |
| | | ES $\uparrow$ | ppl. DD $\downarrow$ | ES $\uparrow$ | acc. DD $\downarrow$ |
| No sharing | $O((m+n)^2 N)$ | **0.86** | **0.195** | **> 0.99** | 0.001 |
| No norm. | $O((m+n)^2)$ | 0.02 | 0.370 | 0.97 | **< 0.001** |
| No ID init. | $O((m+n)^2)$ | 0.27 | 0.898 | 0.94 | **< 0.001** |
| Only $u_\ell$ | $O(m^2)$ | 0.63 | 0.559 | 0.98 | 0.002 |
| Only $\delta_{\ell+1}$ | $O(n^2)$ | 0.80 | 0.445 | **0.99** | 0.001 |
| Only smaller | $O(\min(m,n)^2)$ | 0.80 | 0.593 | 0.98 | 0.002 |
| MEND | $O((m+n)^2)$ | **0.86** | 0.225 | **> 0.99** | 0.001 |

**Table 6: Ablating various properties of MEND** on the Wikitext and zsRE question-answering editing problems. $m = \dim(u_\ell)$, $n = \dim(\delta_{\ell+1})$, and $N$ is the number of layers being edited. Removing MEND's identity initialization and input normalization noticeably lowers editing performance, and relaxations of MEND, particularly the 'only smaller' variant that only outputs pseudoactivations *or* pseudodeltas, whichever is smaller, show competitive performance, which bodes well for scaling MEND to 100 billion+ parameter models.

## 5.4 ABLATIONS & MEND VARIANTS

Table 6 shows ablations of MEND's parameter sharing, identity initialization, and input normalization as well as three variants of MEND that reduce total parameters: only computing pseudoactivations $u_\ell$, only pseudodeltas $\delta_{\ell+1}$, or only whichever of $u_\ell$ or $\delta_{\ell+1}$ is lower-dimensional (layer-dependent for non-square weights). 'No ID init.' replaces zero initialization with Xavier/Glorot initialization (Glorot and Bengio, 2010). Removing *either* input normalization or identity initialization significantly reduces edit effectiveness (and increases training time ~10x). Sharing parameters across model editor networks incurs relatively little performance cost, and editing *only* the smaller of the pseudoactivations and pseudodeltas, the most most lightweight version of MEND, still produces effective edits, suggesting that MEND could scale to even much larger models for which $m + n$ approaches $10^5$ (Brown et al., 2020) but $\min(m, n)$ remains close to $10^4$. Appendix E shows an additional ablation editing attention matrices, rather than MLP weights, finding that editing MLP weights is consistently more effective for large models.

## 6 DISCUSSION

**Conclusion.** We have presented an efficient approach to editing very large (10 billion+ parameter) neural networks, which we call Model Editor Networks with Gradient Decomposition or MEND. We showed that MEND is the only method that successfully edits the largest publicly-available Transformer models from the GPT and T5 model families. To do so, MEND treats the model editing problem itself as a learning problem, using a relatively small edit dataset to learn model editor networks that can correct model errors using only a single input-output pair. MEND leverages the fact that gradients with respect to the fully-connected layers in neural networks are rank-1, enabling a parameter-efficient architecture that represents this gradient transform.

**Limitations & Future Work.** A limitation of existing model editors (including MEND) is the approach to enforcing locality of edits. The failure mode of over-generalization (bottom of Table 2) shows that locality examples (i.e., negative examples) are not challenging enough to prevent the model from sometimes changing its output for distinct but related inputs. Alternative locality losses or harder negative mining may help address this problem. Further, existing language-based editing datasets use backtranslation to evaluate edit generality (and our Wikitext dataset uses a truncation heuristic). Such equivalence neighborhoods do not assess a model's ability to use the knowledge in an edit example to correctly answer questions about other topics whose answer is *implied* by the content of the edit example (e.g., for *Who is the UK PM? Boris Johnson*, does the edited model correctly answer *Is Boris Johnson a private citizen?*). Counterfactual data augmentation (Kaushik et al., 2020) may be useful for constructing richer evaluation cases for edit generality. Future work might also apply MEND to other types of edits, such as reducing the frequency of toxic generations after observing toxic outputs, relabeling entire classes of images from one example, or adjusting a robot's control policy to avoid particular actions, as MEND is not limited to editing transformer models. Finally, MEND's gradient decomposition is not in principle limited to the model editing problem, and it might enable efficient new gradient-based meta-learning algorithms.

## ACKNOWLEDGEMENTS

We gratefully acknowledge Angeliki Lazaridou for insightful early discussions regarding temporal generalization in language models; Spencer Braun for implementing exploratory experiments that motivated this project; Mitchell Wortsman, Gabriel Ilharco, Stephanie Chan, and Archit Sharma for insightful discussions and encouragement; and Michael Chang, Michael Janner, and Ashwin Paranjape for feedback on an early version of the paper. Eric Mitchell gratefully acknowledges the support of a Knight-Hennessy graduate fellowship. Chelsea Finn and Chris Manning are fellows in the CIFAR Learning in Machines and Brains program.

## ETHICS STATEMENT

This work uses large language models pre-trained on text scraped from the internet. These massive training corpora (and therefore the models trained on them) may contain (or produce) content that is counter to the values of the ICLR community. Algorithms for model editing may provide one tool (among others) to mitigate this problem by enabling maintainers of large models to change certain undesirable model behaviors as they are discovered. On the other hand, a model editor could also be used to exacerbate the very model behaviors that we hope to eliminate, depending on who is wielding it. This dual use is a risk for many machine learning technologies. Specifically, effective editing algorithms (including MEND and others) may enable maintainers of deployed neural networks to include backdoors or other planned vulnerabilities/hidden behaviors into their models.

## REPRODUCIBILITY

To foster reproducibility, we have provided a detailed description of the proposed algorithm in Section 3, as well as additional details regarding experimental setup, hyperparameters, and implementations of comparison algorithms in Section C. Our experiments use fixed random seeds for data sampling and model editor initialization, enabling reproducible results. Section C.4 describes how to obtain the pre-existing datasets and models we used in our experiments (from De Cao et al. (2021)). See project website at https://sites.google.com/view/mend-editing for links to code and data.

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

## A  EFFECTIVE INITIALIZATION AND NORMALIZATION FOR MEND NETWORKS

Although random weight initialization is effective in many settings, it sacrifices the prior that the raw fine-tuning gradient is a useful starting point for editing. Our ablations show that it also leads to less effective edits. For this reason, we initialize MEND to the identity function using a residual connection (He et al., 2016) and a partially random, partially zero-initialization strategy related to Fixup (Zhang et al., 2019). Referring back to Eqs. 3a,b, $U_1$ and $U_2$ are initialized with zeros, and $V_1$ and $V_2$ use standard Xavier uniform initialization (Glorot and Bengio, 2010) (also see Figure 2). Beyond the initialization, input scaling also presents a challenge: inputs to a MEND network ($u_\ell$ and $\delta_{\ell+1}$) can differ in magnitude by several orders of magnitude. This poor conditioning causes training to be slow and edit performance to suffer (see Section 5.4). Input normalization addresses this issue; we normalize each dimension of both $u_\ell$ and $\delta_{\ell+1}$. The input to $g_\ell$ is the concatenation of $\bar{u}_\ell = \text{norm}(u_\ell)$ and $\bar{\delta}_{\ell+1} = \text{norm}(\delta_{\ell+1})$, where $\bar{u}_\ell$ and $\bar{\delta}_{\ell+1}$ are normalized to have zero mean and unit variance, with means and variances computed over the edit train set and the sequence index.

## B  EXTENDED DISCUSSION OF RELATED WORK

Model editing shares with continual learning (McCloskey and Cohen, 1989; Parisi et al., 2019) the goal of assimilating or updating a model's behavior without forgetting old information or behaviors, commonly known as the problem of catastrophic forgetting (McCloskey and Cohen, 1989; Ratcliff, 1990; Kirkpatrick et al., 2017). However, in continual learning settings, a model is typically expected to learn wholly new behaviors or datasets (Kirkpatrick et al., 2017; Parisi et al., 2019) without forgetting, while in this work we consider more localized model edits. Further, continual learning generally considers long sequences of model updates with minimal memory overhead, while our work generally considers an edit or batch of edits applied all at once.

Additionally, min-norm parameter fine-tuning has also been considered in past work in the context of editing (Zhu et al., 2020) and traditional model fine-tuning (Guo et al., 2021), where the parameters of the edited or fine-tuned model $\theta'$ are penalized (or constrained) from drifting too far from the original model parameters $\theta$ using various norms, including L0, L2, and L-$\infty$. While min-norm constraints may be an effective regularization for traditional fine-tuning settings where fine-tuning data is abundant, the experiments conducted in De Cao et al. (2021) show that parameter-space norm constraints are insufficient constraints to prevent significant model degradation when fine-tuning on a single edit example.

### B.1  EDITABLE NEURAL NETWORKS (ENN)

Editable neural networks (Sinitsin et al., 2020) search for a set of model parameters that both provide good performance for a 'base task' (e.g., image classification or machine translation) and enable rapid editing by gradient descent to update the model's predictions for a set of 'edit examples' without changing the model's behavior for unrelated inputs. ENN optimizes the following objective, based on the MAML algorithm (Finn et al., 2017):

$$\mathcal{L}_{\text{ENN}}(\theta, \mathcal{D}_{\text{base}}, \mathcal{D}_{\text{edit}}, \mathcal{D}_{\text{loc}}) = L_{\text{base}}(\mathcal{D}_{\text{base}}, \theta) + c_{\text{edit}} \cdot L_{\text{edit}}(\mathcal{D}_{\text{edit}}, \theta') + c_{\text{loc}} \cdot L_{\text{loc}}(\mathcal{D}_{\text{loc}}, \theta, \theta'). \quad (5)$$

The first term of Equation 5 is the base task loss; for a generative language model, we have $L_{\text{base}}(\mathcal{D}_{\text{base}}, \theta) = -\log p_\theta(\mathcal{D}_{\text{base}})$ where $\mathcal{D}_{\text{base}}$ is a batch of training sequences. $L_{\text{base}}$ is the edit *reliability* loss, encouraging the model to significantly change its output for the edit examples in $\mathcal{D}_{\text{edit}}$. Finally, $L_{\text{loc}}$ is the edit *locality* loss, which penalizes the edited model $\theta'$ for deviating from the predictions of the pre-edit model $\theta$ on $\mathcal{D}_{\text{loc}}$, data unrelated to $\mathcal{D}_{\text{edit}}$ and sampled from the same distribution as $\mathcal{D}_{\text{base}}$. See Sinitsin et al. (2020) for a more detailed explanation of ENN training and alternative objectives for $L_{\text{edit}}$ and $L_{\text{loc}}$.

**Comparing ENN and MEND.** The key conceptual distinction between ENN and MEND is that ENN encodes editability into the parameters of the model itself (*intrinsic editability*), while MEND provides editability through a set of learned parameters that are independent from the model parameters (*extrinsic editability*). An advantage of ENN is that no new parameters are added in order to provide editability. However, this approach comes with several drawbacks. First, the MAML-based objective ENN optimizes is expensive, particularly in terms of memory consumption (see Figure 4). By further training the model parameters themselves, ENN cannot guarantee that the editable model it produces will make the same predictions as the original model. In order

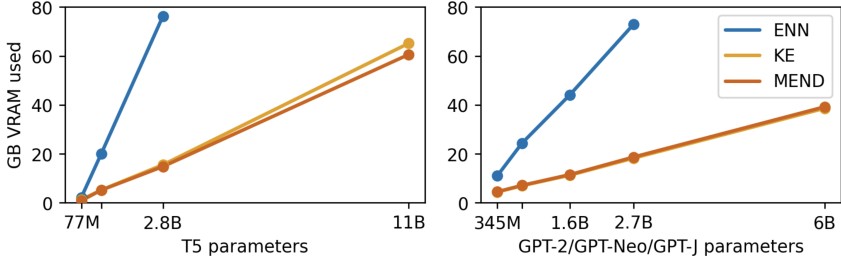

**Figure 4: GPU VRAM consumption** for training MEND, KE, and ENN in float32. MEND and KE's memory consumption remain tractable for a single GPU (using 2×bfloat16 memory usage (Wang and Kanwar, 2019) for T5-11B), while ENN's memory usage increases much more rapidly, making it impractical to run on a single GPU. Values are computed without gradient checkpointing. Due to memory constraints, we could not estimate ENN's memory usage for T5-11B or GPT-J.

to approximately enforce this constraint during training, ENN must use an extra copy of the original base model to ensure that the editable model's predictive distribution does not differ too much from it. This incurs significant additional memory costs, particularly when training ENN for very large models, for which the parameters of the model alone occupy a significant amount of VRAM. Another cause for the significant VRAM consumption of ENN is the need to compute activations and gradients for the model parameters; even if we edit only the last layer, ENN trains the rest of the model so that the last layer gradient is productive, requiring activations and gradients to be computed for the entire model. On the other hand, extrinsic editors like MEND and KE do not require updating the base model itself, thereby computing gradients for far fewer parameters. Future work might investigate approaches to reducing the memory consumption of ENN, although the requirement to retain a copy of the original model in order to enforce locality creates a relatively high lower bound on the amount of memory that ENN might use.

Regardless of memory consumption, extrinsic editors have the potential advantage of being able to edit more than one model; in theory, we might amortize the cost of training MEND over several base models at once. On the other hand, intrinsic editability must by definition be re-learned separately for each base model.

## B.2 KNOWLEDGEEDITOR (KE)

De Cao et al. (2021) propose KNOWLEDGEEDITOR, a hypernetwork-based approach for editing the knowledge in language models. KE is an RNN that conditions explicitly on the input, incorrect output, and new desired label and outputs a mask $m_i$, offset $b_i$, and a scalar scaling factor $\alpha$ to the gradient $\nabla_{W_i}$ for several of the weight matrices in a transformer model, where $m_i, b_i, \nabla_{W_i} \in \mathbb{R}^{d \times d}$ for a $d \times d$ weight matrix. The update to the model is $\theta' = \theta - \alpha(m_i \odot \nabla_{W_i}) + b_i$. Because the weight matrices in state-of-the-art transformer models are very high-dimensional, the mask and offset output by KE are rank-1 to retain tractability.

**Comparing KE and MEND.** KE more closely resembles MEND in that it is also an extrinsic model editor. However, while MEND directly maps model gradients into model edits, the KE model editor uses the raw edit example as an input, outputting a single rank-1 mask and rank-1 offset over the fine-tuning gradient. We hypothesize that the KE model faces several challenges that MEND avoids. First, mapping the edit example itself into a model updates requires a translation from the high-level modality of data examples into the very low-level modality of model parameter updates. Solving this translation requires making additional design decisions (e.g., how to feed the edit input and label into the editor, what architecture to use for the editor), the optimal design for which may vary across problems. Further, by not conditioning directly on the gradient, KE forgoes a rich source of information about which parameters of the model are most responsible for updating the model's outputs. In addition, by operating on the token-wise activations and gradients (i.e., the gradients are not summed over the sequence/batch, but are kept as per-sequence element activation and gradient vectors), MEND outputs a rank-1 model edit for each token in the input and output sequence. The final output of MEND is the sum of these, which has rank of order 10 or even 100, depending on the problem. In contrast, the KE editor outputs only a rank-1 gradient mask and rank-1 gradient offset, regardless of the information content of the edit example. This rank-1 constraint, irrespective of the size of the input, which we hypothesize causes KE's failure to perform well for the Wikitext editing

| $x_\text{e}, y_\text{e}$ | Nepal borders France. **Yes** |
|---|---|
| $x_\text{loc}$ | Belgium is made up of three regions. |
| $x'_\text{e}, y'_\text{e}$ | Nepal is bordered by France. **Yes** |

**(a) FEVER fact-checking editing dataset example.** In this case, the locality loss is computed as the KL divergence between the Bernoulli distribution produced by the pre-edit and post-edit model for the locality example $x_\text{loc}$.

| $x_\text{e}$ | Which continent is Mount Andrews on? **South America** |
|---|---|
| $x_\text{loc}, y_\text{loc}$ | To which fictional work does Dennis Rickman belong in? **EastEnders** |
| $x'_\text{e}, y'_\text{e}$ | In which continent is Mount Andrews located? **South America** |

**(b) zsRE question-answering editing dataset example.** Because computing the KL divergence of the model over all possible answers to the question is computationally expensive, we use the label (EastEnders) and compute the KL divergence between the pre- and post-edit model at each of these tokens as an approximation.

**Table 7: Editing data samples** from the FEVER fact-checking and zsRE question-answering editing datasets from De Cao et al. (2021). **Bold text** corresponds to labels used for editing or approximating the locality constraint.

task, which has significantly higher information content labels (10 tokens) than the FEVER or zsRE tasks.

## C    EXPERIMENTAL DETAILS

For GPT and BERT-style models, all experiments edit the MLP weights in the last 3 transformer blocks (6 weight matrices total). For BART and T5-style models, all experiments edit the MLP weights in the last 2 transformer blocks in both the encoder and the decoder (8 weight matrices total). We found that editing MLP layers generally provides better editing performance (across algorithms) than editing attention layers. In line with past work (De Cao et al., 2021), all reported performance numbers are on the validation set. For all algorithms, we use early stopping to end training early if the validation loss $L = c_\text{edit}L_\text{e} + L_\text{loc}$) does not decrease for 20000 steps on a subset of 500 validation examples, with a maximum number of training steps of 500,000. We use a batch size of 10 (with gradient accumulation) and the seed 0 for all experiments. Tables 7 and 8 show examples from each dataset used in our experiments.

### C.1    HYPERPARAMETERS

**Fine-tuning.** The fine-tuning baselines use model-dependent learning rates, which we found important in achieving good fine-tuning performance; using too large of a learning rate causes decreased locality (increased model degradation), while a learning rate too small causes slow edits. We use edit learning rates of 5e-6 for GPT-Neo and GPT-J and 1e-4 for T5 models, and 1e-6 for the smaller models, aiming to complete edits in less than 100 fine-tuning steps (as in De Cao et al. (2021)). For the fine-tuning + KL-constraint baseline, we fine-tune on the loss $c_\text{edit}L_\text{e} + L_\text{loc}$, using a smaller $c_\text{edit}$ than for the learned algorithms (1e-2 for all models except GPT-J, which required 1e-3). Larger values of $c_\text{edit}$ provide little benefit from the locality loss. To compute $L_\text{loc}$, we use a batch size of one new example $x_\text{loc}$ from the full edit training set $D^{tr}_{edit}$ at each time step.

**ENN.** We use an initial inner loop learning rate of 1e-2, but allow this value to be learned in the outer loop, which we find improves performance over the fixed inner loop learning rate version in Sinitsin et al. (2020). For all experiments, ENN fine-tunes all model parameters during training (even when we only edit the last few layers). We also use only a single inner loop update step for computational reasons, which differs from the multi-step version used for the smaller models used by Sinitsin et al. (2020). Our edit loss is also a slight simplification of the edit loss used by Sinitsin et al. (2020), which is

$$l_e(\theta) = -\log p_\theta(y_e|x_e, \theta) + \max_{y_i} \log p_\theta(y_i|x_e, \theta) \tag{6}$$

The first term of this loss is the edit loss we use in our work; the second term is primarily intended to provide the property that $l_e(\theta) \leq 0$ when an edit is successful so that the iterative editing process can be stopped. However, in this work, because we use only a single gradient step of editing for

ENN, this property is less important, and the second term simply amounts to an additional emphasis on pushing down specifically the largest incorrect logit (which the first term already does implicitly).

**KE**    We use the implementation of KE provided by De Cao et al. (2021), which can be found at https://github.com/nicola-decao/KnowledgeEditor, with minor changes to the computation of the KL constraint for consistency with other algorithms (see below). We use a learning rate of 1e-5.

## C.2    Computing the locality constraint

Computing the true KL-divergence between the pre- and post-edit model $\text{KL}(p_\theta(\cdot|x_{\text{loc}})\|p_{\theta'}(\cdot|x_{\text{loc}}))$ quickly becomes computationally prohibitive for model outputs of more than a few tokens, requiring marginalization over possible answers. We therefore approximate this KL-divergence using samples from the dataset.[6] For the seq2seq question-answering problem, we evaluate the KL divergence only at the tokens of the answer $y_{\text{loc}}$, giving $\text{KL}_{\text{approx}}^{\text{seq2seq}}(\theta, \theta') = \frac{1}{|y_{\text{loc}}|}\sum_{i=1}^{|y_{\text{loc}}|}\text{KL}(p_\theta(\cdot|x_{\text{loc}}, y_{\text{loc}}^{<i})\|p_{\theta'}(\cdot|x_{\text{loc}}, y_{\text{loc}}^{<i}))$, where $p(\cdot|x_{\text{loc}}, y_{\text{loc}}^{<i})$ is the distribution over next tokens $y_i$ given the locality input $x_{\text{loc}}$ and the label tokens for previous timesteps $y_{\text{loc}}^{<i}$. Similarly, for the Wikitext setting, we define $\text{KL}_{\text{approx}}^{\text{auto}}(\theta, \theta') = \frac{1}{|x_{\text{loc}}|}\sum_{i=1}^{|x_{\text{loc}}|}\text{KL}(p_\theta(\cdot|x_{\text{loc}}^{<i})\|p_{\theta'}(\cdot|x_{\text{loc}}^{<i}))$. For FEVER fact-checking we compute the exact KL-divergence between Bernoulli distributions in closed form.

## C.3    Environment Details

All runs are trained entirely on a single NVIDIA RTX Titan or A40 GPU. No gradient checkpointing or memory-reduction optimizations are used, although bfloat16 is used to fit the largest T5 model onto our GPU. In full precision, the parameters alone of the T5-11B model use all of the memory of our largest GPU. VRAM consumption for training MEND and KE on T5-11B (Figs. 3 and 4) is estimated by doubling the bfloat16 VRAM usage (Wang and Kanwar, 2019). While doubling half precision enabled estimating the memory consumption of ENN, we were unable to train ENN in half precision without numerical instability. All models are based on Huggingface Transformers implementations (Wolf et al., 2019) with some modifications in line with De Cao et al. (2021). We use PyTorch (Paszke et al., 2019) for all experiments, specifically using the Higher library (Grefenstette et al., 2019) in order to implement the bi-level optimization in ENN as well as the inner loop of model editing for all algorithms.

## C.4    Dataset Construction & Examples

Datasets are constructed to provide pairs of edit input $x_{\text{e}}$ and plausible edit label $y_{\text{e}}$. The edit label is not necessarily the 'correct' label; the goal is to provide realistic instances of the *types* of data we would expect to see during test. For example, our dataset might have a sample such as $x_{\text{e}}$ = *Where was Ursula K. Le Guin born?* and $y_{\text{e}}$ = *Addis Ababa, Oromia, Ethiopia*, even though Ursula K. Le Guin was born in Berkeley, California, USA. However, this fictitious example is still a useful assessment of our model's ability to perform the general type of edit of 'change a person's birthplace'. For the zsRE question-answering dataset De Cao et al. (2021) generate fictitious $y_{\text{e}}$ in this manner using the top predictions of a BART model fine-tuned on the task of question answering followed by manual human filtering. In practice, this produces alternate edit labels that are plausible and whose types match with the original label. For FEVER fact-checking, there are only two choices for labels, and we sample edit targets 1 and 0 with equal probability. For Wikitext generation, we use a distilGPT-2 model to generate plausible 10-token continuations for a given Wikitext prefix, with the similar motivation to zsRE of providing edit targets that share the structure of the types of edits that we will apply in practice, even if they are not always factual. When qualitatively assessing MEND to correct real errors of the base model using the factual labels, we find that MEND performs reliably, indicating that these label generators provide reasonable proxies for 'real' model edits.

---

[6]We justify this choice by the fact that the model's predictive distribution is similar to the locality sample distribution (as locality samples are drawn from the dataset the model was originally trained on). While this is not as principled as a true Monte Carlo estimate using samples from the model itself, it is reduces computational requirements of training and is easier to implement; the generally low drawdown for most models indicates that this approximation still provides a good locality constraint in practice.

| $x_\mathrm{e}, y_\mathrm{e}$ | Saprang was considered one of the top contenders to lead the army and the junta after CNS leader Sonthi Boonyaratkalin's mandatory retirement in 2007. However, in September 2007 he was demoted to be Deputy Permanent Secretary of the Defense Ministry, while his rival, General Anupong Paochinda, was promoted **to Deputy Attorney General. Later, he was replaced** |
|---|---|
| $x_\mathrm{loc}$ | In 1663 Scottish mathematician James Gregory had suggested in his Optica Promota that observations of a transit of the planet Mercury, at widely spaced points on the surface of the Earth, could be used to calculate the solar parallax and hence the astronomical unit using triangulation. Aware of this, a young Edmond Halley made observations of such a transit on 28 October O.S. 1677 from Saint Helena but was disappointed to find that only Richard Towneley in Burnley, Lancashire had made another accurate observation of the event whilst Gallet, at Avignon, simply recorded that it had occurred. Halley was not satisfied that the resulting calculation of the solar parallax at 45 " was accurate. |
| $x'_\mathrm{e}, y'_\mathrm{e}$ | However, in September 2007 he was demoted to be Deputy Permanent Secretary of the Defense Ministry, while his rival, General Anupong Paochinda, was promoted **to Deputy Attorney General. Later, he was replaced** |

**Table 8: Training set example from the Wikitext editing dataset.** Bolded text corresponds to the edit labels $y_\mathrm{e}$ and $y'_\mathrm{e}$. The locality example $x_\mathrm{loc}$ is used to constrain the pre- and post-edit model's predictive distributions to be similar at for *every* token in the sequence.

## D    RANK-1 GRADIENT FOR MLPS

In the simplified case of an MLP and a batch size of 1, we describe the rank-1 gradient of the loss $L$ with respect to the layer $\ell$ weight matrix $W_\ell$. We define the inputs to layer $\ell$ as $u_\ell$ and the *pre-activation* inputs to layer $\ell + 1$ as $z_{\ell+1} = W_\ell u_\ell$. We define $\delta_{\ell+1}$ as the gradient of $L$ with respect to $z_{\ell+1}$ (we assume that $\delta_{\ell+1}$ is pre-computed, as a result of standard backpropagation). We will show that the gradient of the loss $L$ with respect to $W_\ell$ is equal to $\delta_{\ell+1} u_\ell^\top$.

By the chain rule, the derivative of the loss with respect to weight $W_\ell^{ij}$ is equal to

$$\frac{\partial L}{\partial W_\ell^{ij}} = \sum_k \frac{\partial L}{\partial z_{\ell+1}^k} \frac{\partial z_{\ell+1}^k}{\partial W_\ell^{ij}} = \frac{\partial L}{\partial z_{\ell+1}^i} \frac{\partial z_{\ell+1}^i}{\partial W_\ell^{ij}} \tag{7}$$

the product of the derivative of $L$ with respect to next-layer pre-activations $z_{\ell+1}^i$ and the derivative of next-layer pre-activations $z_{\ell+1}^i$ with respect to $W_{ij}$. The second equality is due to the fact that $\frac{\partial z_{\ell+1}^k}{\partial W_\ell^{ij}} = 0$ for $k \neq i$. Noting that $z_{\ell+1}^i = \sum_j u_\ell^j W_\ell^{ij}$, we can replace $\frac{\partial z_{\ell+1}^i}{\partial W_\ell^{ij}}$ with simply $u_\ell^j$ in Equation 7. Further, we defined $\delta_{\ell+1}$ to be exactly $\frac{\partial L}{\partial z_{\ell+1}^i}$. Making these two substitutions, we have

$$\frac{\partial L}{\partial W_\ell^{ij}} = \delta_{\ell+1}^i u_\ell^j \tag{8}$$

or, in vector notation, $\nabla_{W_\ell} L = \delta_{\ell+1} u_\ell^\top$, which is the original identity we set out to prove.

## E    EDITING ATTENTION PARAMETERS

Our experiments edit weights in the MLP layers of large transformers. Here, Table 9 shows the results of editing the attention layers, rather than MLP layers, observing that editing attention layers generally leads to reduced performance compared to editing MLP layers. For this comparison, we edit the same transformer blocks as for our main editing experiment in Table 3, but we edit the query/key/value/output matrices for each block instead of the two MLP matrices. The observation that editing MLP layers is more effective generally aligns with past work (Geva et al., 2021) suggesting that the MLP layers in Transformer architectures store human-interpretable, high-level concepts in the later layers of the model, motivating our choice of editing these layers in our original experiments. Further, we hypothesize that the improved effectiveness of editing MLP layers may simply be based on the fact that they make up a large majority of model parameters, as the MLP hidden state is often much higher-dimensional than the model's hidden state.

| | Wikitext Generation | | | | zsRE Question-Answering | | | |
| --- | --- | --- | --- | --- | --- | --- | --- | --- |
| | GPT-Neo (2.7B) | | GPT-J (6B) | | T5-XL (2.8B) | | T5-XXL (11B) | |
| Editor | ES ↑ | ppl. DD ↓ | ES ↑ | ppl. DD ↓ | ES ↑ | acc. DD ↓ | ES ↑ | acc. DD ↓ |
| MEND-attention | 0.73 | 0.068 | 0.54 | 0.122 | 0.63 | 0.001 | 0.78 | < 0.001 |
| MEND-mlp (Tab. 3) | **0.81** | **0.057** | **0.88** | **0.031** | **0.88** | 0.001 | **0.89** | < 0.001 |

**Table 9: Editing attention matrices** rather than MLP/feedforward parameters for the models considered in Table 3. Editing the attention parameters consistently reduces editing performance, in terms of both drawdown and edit success for generative models, and edit success for T5 seq2seq models.

| Input | Pre-Edit Output | Edit Target | Post-Edit Output |
| --- | --- | --- | --- |
| 1a: **Who is the president of the USA?** | Donald Trump ✗ | **Joe Biden** | Joe Biden ✓ |
| 1b: Who is the US president? | David Rice Atchison ✗ | - | Joe Biden ✓ |
| 1c: Who is the president of France? | Emmanuel Macron ✓ | - | Emmanuel Macron ✓ |
| 2a: **Who designed the Burj Khalifa?** | British architect Herbert Baker ✗ | **Skidmore, Owings & Merrill** | Skidmore, Owings & Merrill ✓ |
| 2b: Who designed the Eiffel Tower? | Alexandre Gustave Eiffel ✓ | - | Alexandre Gustave Eiffel ✓ |
| 2c: Who designed the Empire State Building? | Shreve, Lamb and Harmon ✓ | - | Shreve, Lamb and Harmon ✓ |
| 2d: Who designed the Sydney Opera House? | Jrn Oberg Utzon ✓ | - | Jrn Oberg Utzon* ✓ |
| 2e: What firm was behind the design for the Burj Khalifa? | McKim, Mead & White ✗ | - | Skidmore, Owings & Merrill ✓ |
| 2f: What firm did the Burj Khalifa? | Jumeirah Group ✗ | - | Jumeirah Group ✗ |
| 3a: **What car company makes the Astra?** | Mahindra ✗ | **Opel** | Opel ✓ |
| 3b: What car company makes the Mustang? | Ford ✓ | - | Ford ✓ |
| 3c: What car company makes the Model S? | Tesla Motors ✓ | - | Tesla ✓ |
| 3d: What car company makes the Wrangler? | Jeep ✓ | - | Jeep ✓ |
| 3e: What car company makes the F-150? | Ford ✓ | - | Opel ✗ |
| 3f: What car company makes the Golf? | Volkswagen AG ✓ | - | Opel ✗ |
| 4a: **What artist recorded Thriller?** | Madonna ✗ | **Michael Jackson** | Michael Jackson ✓ |
| 4b: What artist recorded Dark Side of the Moon? | Pink Floyd ✓ | - | Pink Floyd ✓ |
| 4c: What artist recorded Bridge over Troubled Water? | Simon & Garfunkel ✓ | - | Simon & Garfunkel ✓ |
| 4d: What artist recorded Hotel California? | Don Henley ? | - | Don Henley ? |
| 4e: What band recorded Back in Black? | AC/DC ✓ | - | Michael Jackson ✗ |

**Table 10: Additional examples of using MEND** to edit a 770M parameter T5-large model fine-tuned on Natural Questions (NQ; Kwiatkowski et al. (2019)). Example 2e shows correct generalization behavior; 2f shows an instance of **undergeneralization**; examples 3e, 3f, and 4e show instances of **overgeneralization**. *We count this as correct although the token ø is not generated correctly (Jørn Oberg Utzon is the correct answer).

| | FEVER | | zsRE | | zsRE-hard | | Wikitext | |
|---|---|---|---|---|---|---|---|---|
| | BERT-base | | BART-base | | BART-base | | distilGPT-2 | |
| Editor | ES $\uparrow$ | acc. DD $\downarrow$ | ES $\uparrow$ | acc. DD $\downarrow$ | ES $\uparrow$ | ppl. DD $\downarrow$ | ES $\uparrow$ | ppl. DD $\downarrow$ |
| MEND | **>0.99** | **<0.001** | 0.98 | **0.002** | **0.66** | **<0.001** | **0.86** | 0.225 |
| Cache ($\epsilon^*$) | 0.96 | **<0.001** | **>0.99** | **0.002** | 0.32 | 0.002 | 0.001 | **0.211** |
| Cache ($\frac{1}{2}\epsilon^*$) | 0.70 | <0.001 | 0.70 | <0.001 | – | – | <0.001 | 0.037 |
| Cache ($2\epsilon^*$) | >0.99 | 0.250 | 1.00 | 0.220 | – | – | 0.002 | 2.770 |

**Table 11: Comparing MEND with a caching-based approach to editing.** For purposes of the comparison, the caching hidden-state similarity threshold $\epsilon^*$ is the one that gives similar drawdown to MEND. We found $\epsilon^*$ to be $6.5, 3, 2.5$ for FEVER, zsRE, and Wikitext, respectively. **Top half.** Caching gives slightly better performance for zsRE, slightly worse performance for FEVER, and total failure for Wikitext editing, likely owing to the longer, more complex contexts in the Wikitext data. **Bottom half.** Caching is relatively sensitive to the chosen threshold, which needs to be tuned separately for each new task.

## F    ADDITIONAL QUALITATIVE EXAMPLES OF MEND

We provide additional qualitative examples of using MEND to edit a larger 770M parameter T5-large model (Roberts et al., 2020) in Table 10. These examples include an instance of **undergeneralization**, in which the edit example's output is correctly edited, but other examples in the equivalence neighborhood of the edit example do not change (see 2f in Table 10)). In addition, we highlight the failure case of **overgeneralization**, in which the model's post-edit output for superficially similar but semantically distinct inputs is also the edit target; for example 3e, 3f, and 4e in Table 10. Mitigating these failure cases for model editors (ensuring is an important priority for future work,

## G    EDITING THROUGH CACHING

Another simple approach to editing might be to cache the final layer hidden state $z_e$ (averaged over the sequence length) of the edit example $x_e$ and the tokens of the corresponding edit label $y_e$. After an edit is performed, if the model receives a new input $x$ whose final layer hidden state $z$ is close to $z_e$ (i.e. $\|z - z_e\|_2 < \epsilon$), then the model outputs $y_e$ instead of its normal prediction. Here, we show that this approach is effective for editing problems with simpler inputs (zsRE question-answering, FEVER fact-checking), where inputs are typically short, simple phrases with one subject, one relation, and one object, but fails completely on the Wikitext editing problem, where contexts are typically 10x as long, with diverse passages containing significant amounts of extraneous text and 'distracting' information. The results are presented in Table 11. We include the 'optimal' threshold $\epsilon^*$ (the threshold that achieves similar drawdown to MEND), as well as the result of using $2\epsilon^*$ and $\frac{1}{2}\epsilon^*$. We observe that the caching approach is fairly sensitive to the threshold hyperparameter, and a threshold that works well for one task may not work well for others.

For zsRE question answering, $z$ is computed as the average hidden state of the question tokens; for FEVER fact-checking, $z$ is the average hidden state of the fact statement tokens. For generative modeling, when predicting the token at time step $t$, we compute $z_t$ as the average hidden state for all previously seen tokens $< t$. In order to compute perplexity for the caching approach, we output one-hot logits corresponding to $y_e$. We experimented with scaling the one-hot logit by different factors, but found scaling by 1 to work well; scaling corresponds to changing the model's confidence in its edit prediction but doesn't change the prediction itself or the edit success.

