# OpenReview forum: "Fast Model Editing at Scale"
_ICLR.cc/2022/Conference — ICLR 2022 Poster_

### Official Review · Reviewer_vuAW · 2021-10-29

**Correctness:** 4
**Technical Novelty And Significance:** 3
**Empirical Novelty And Significance:** 3
**Recommendation:** 8
**Confidence:** 4

**Main Review:**

The problem of editing models accurately with minimal side effects is an important one, and I believe this paper presents an elegant and effective method for doing so that will also be of great interest to the audience at ICLR 2022.

As the authors have correctly pointed out, existing model editing techniques have strong limitations that discourage them from being used at scale. MEND breaks free of several of these limitations, which makes this method particularly appealing:

- Edits require only a "single example illustrating the model's error" (where an "example" is composed of the edit example, equivalence examples, and a locality example). There is no need for the original training dataset
- There are no modifications to the original weights until the edit occurs. This makes the MEND algorithm lightweight, and possibly patchable (* see "Questions for the authors").
- The edit success is higher than that of strictly fine-tuning of the model, while also regularly causing less damage to existing predictions than smart fine-tuning would.
- The modifications are resource-efficient, requiring only first-order gradient calculations, and can be trained quickly with low GPU resources (* see Feedback)

Unfortunately, results in Table 4 show that MEND fails to perform better than ENN on smaller models on the task of generation, and as such I see this method being most helpful for the large model domain they are targeting.

**Feedback**

1. The claim of editing very large networks (6B+ parameters) is a bit deceitful — in the experiments, it seems the authors only edited the final 3 MLP layers of the Transformer. Please qualify the claims of speed (e.g., "MEND can be trained on a single GPU in less than a day even for 10 billion+ parameter models") somewhere in the paper, as not all of these parameters are being updated. This statement requires that important gradients all reside in a small subset of total model parameters.
2. Sec 5 "Comparison of Model Editors": It is not clear if the **fine-tune** and **fine-tune + KL** models operate on the same subset of parameters as MEND or on all the parameters of a Network. Similarly with **KE**. Please clarify.
3. There is no explicit comment of what FT, FT+KL mean in Tables 1 & 3 of the paper (though their descriptions appear in section "Comparison of Model Editors"). Please add, possibly in the caption itself.
4. Minor: I feel the paper is lacking qualitative examples, all of which are contained in Table 2.  There is only one example of drawdown post MEND edit (Table 2 input 3b), and in this case it was not harmful. I would have appreciated additional qualitative "before and after" examples in the appendix, as this is the main message of the paper.
5. Q: I was surprised to find discussion on ["Parameter-Efficient Transfer Learning with Diff Pruning"](https://arxiv.org/pdf/2012.07463.pdf) by Guo et al. 2020 to be absent from the paper. While MEND has several advantages over this technique (i.e., MEND reportedly achieves accuracy *higher* than traditional fine-tuning), a perk of the Diff Pruning technique (as well as "Knowledge Neuron" editing by Dai et al. 2021) is the ability for each edit to be viewed as a "patch" completely independent from the model parameters themselves. Then, knowledge changes can be added/subtracted from the model independently. Is this a reasonable expectation to have of MEND? If so, this could be an additional impact point of the paper. Even these can be viewed as "patches" that are applied to the original model's weights in terms of the $\alpha \tilde{\nabla}_{W_\ell}$ in the equation

$$\tilde{W} = W_\ell - \alpha \tilde{\nabla}_{W_\ell}$$

6. Q: In A.2p17, the authors mention that they attempted to edit attention weights but that the results were not as effective as edits on the MLPs. I would be interested to see those results, especially as more papers are coming out indicating the different roles and impacts performed by the MLP layers vs the Attention layers.

The paper overall is well written, and this reviewer did not notice grammatical mistakes.

**Summary Of The Paper:**

This paper tackles the challenging model of editing a potentially very large pre-trained model in a way that is specific (local) yet comprehensive (general). Their method, called MEND, take as input the decomposed, information-rich gradient from fine-tuning to learn the parameters for their meta-networks that subsequently perform the updates. The authors show that it works better than existing model editing techniques on large models like GPT-Neo, GPT-J, T5-XL, T5-XXL, and some smaller models like BERT-base, and distilGPT-2.

**Summary Of The Review:**

The paper proposes an effective, novel, and general technique for manually editing pre-trained models. While the method is not universally better than other editing techniques (especially for smaller language models), I can see immediate interest and impact of this work in large AI models. I vote to accept the paper.

---

> ### Author Response · Authors · 2021-11-16
> **Response to reviewer vuAW**
>
> Thank you for your comments! We're happy to hear you felt our work would be of great interest to the ICLR community, and that MEND is an elegant approach to the editing problem. To address your comments:
>
> > MEND fails to perform better than ENN on smaller models on the task of generation, and as such I see this method being most helpful for the large model domain they are targeting.
>
> To highlight another sense in which MEND is a scalable model editor, we've added ENN's performance to the batched editing ablation in Table 5 (using a "small" BART-base model). MEND significantly outperforms ENN when scaling the number of simultaneous edits, rather than model size:
>
> | Edits | ENN ES | MEND ES | ENN DD | MEND DD |
> |-------|-----------|------------|-------|-----------|
> |5|0.94|0.97|0.007|0.005|
> |25|0.35|0.96|0.005|0.008|
> |125|0.11|0.67|0.006|0.012|
>
>
> > The claim of editing very large networks (6B+ parameters) is a bit deceitful
>
> We apologize for using wording that could mislead readers; this was sincerely not our intent. We have clarified in the abstract and paper that we are editing the **behavior** of very large networks. To make it more clear that MEND is not editing all model parameters themselves, we have also clarified in Section 5 (footnote 3) the specific (subset of) parameters edited for each model/experiment.
>
> >  It is not clear if the fine-tune and fine-tune + KL models operate on the same subset of parameters
>
> We have clarified in Section 5 that for each dataset, all editing algorithms edit the same model parameters to ensure a fair comparison (this was originally only mentioned in Appendix B).
>
> > There is no explicit comment of what FT, FT+KL mean in Tables 1 & 3 of the paper
>
> Thank you for this suggestion- we have added an additional description of these methods in the respective captions.
>
> > I would have appreciated additional qualitative "before and after" examples in the appendix
>
> We added Appendix F (Table 10) to the paper, which includes more qualitative examples of both success and failure cases of MEND.
>
> > I was surprised to find discussion on "Parameter-Efficient Transfer Learning with Diff Pruning" by Guo et al. 2020 to be absent from the paper
>
> Thank you for this suggestion; we have added discussion of this paper to the extended related work. We note that de Cao et al. 2021 show that a min-norm parameter-space constraint is not generally strong enough to provide effective edits when only a single example is available, comparing with min-norm fine-tuning approaches based on L2 and L-\infty norms (Modifying Memories in Transformer Models, Zhu et al. 2020). The diff pruning paper also considers transfer learning to new datasets where fine-tuning data is abundant, unlike our editing setting, where we have only a single edit example at test time.
>
> > "Knowledge Neuron" editing by Dai et al. 2021
>
> Dai et al. 2021 present a very interesting approach to editing (and is already cited in the related work section). However, it is not comparable to the methods considered in our experiments because knowledge neuron editing requires a collection of different expressions of the fact to be edited at test time (10 for their experiments) in order to perform attribution and editing, whereas our problem setting considers edits from a single example and label.
>
> > knowledge changes can be added/subtracted from the model independently. Is this a reasonable expectation to have of MEND? If so, this could be an additional impact point of the paper
>
> Yes, patches for different edit examples can be seen as completely independent patches, and can be mixed and matched for a particular model! For the batched editing experiments (see Table 5), the parameter edit for the entire batch is the simple sum of the parameter updates for each edit example, which are computed entirely independently. We agree this is an intriguing property of MEND, which we have emphasized in the revised batched editing experiment section (5.3).
>
> > the authors mention that they attempted to edit attention weights but that the results were not as effective as edits on the MLPs
>
> We've added a new Appendix E that includes the results of editing attention weights instead of MLP layers; editing performance is consistently reduced across models. We suspect that this finding is related both to the observations in Geva et al. (2021) Transformer Feed-Forward Layers Are Key-Value Memories about key-value memories stored in MLP layers, as well as the simple fact that the MLP parameters make up the majority of the model parameters for large transformers. We have included the results here (formatted as edit success/drawdown for each variant/model combination):
>
> | Variant | GPT-Neo | GPT-J | T5-XL | T5-XXL |
> |---|---|---|---|---|
> |MEND-attention|0.73/0.068|0.54/0.122|0.63/0.001|0.78/<0.001|
> |MEND-mlp|0.81/0.057|0.88/0.031|0.88/0.001|0.89/<0.001|
>
> Please let us know if you have any lingering questions, comments, or concerns about the paper!

---

> > ### Comment · Reviewer_vuAW · 2021-11-16
> > **Looks good**
> >
> > Thank you for your thorough responses to my feedback and that of the other reviewers. It seems that all of my initial concerns have been addressed, and it is very good to hear that another reviewer has successfully tried your code. The changes look good, and I believe it makes a compelling paper. I vote accept.

---

### Official Review · Reviewer_5ZVS · 2021-11-01

**Correctness:** 3
**Technical Novelty And Significance:** 3
**Empirical Novelty And Significance:** 4
**Recommendation:** 8
**Confidence:** 4

**Main Review:**

## Practical significance

I would argue that the main merit of this paper is in how it extends the range of **practically useful** applications for model editing.

Being an industrial researcher, I found that editing small models was often not justified, as those models could be re-trained from scratch in a few hours. In turn, editing large models was impractical without large-scale model parallelism. As a result, methods such as Editable Training (referred as ENN) would only make practical sense if one needs to edit a (small) model on device, where re-training is infeasible. However, such use cases are very rare, as companies typically hesitate to let each individual user have its own version of a model on-device, which would complicate debugging and quality assurance.

In contrast, MEND could make model editing much more widespread since (1) MEND can be used for models that cannot be simply re-trained (I was able to run MEND for a GPT2-like model with 41B parameters using a 4x A100-80GB pod with tensor parallelism; that model was trained on 128 GPUs for over a month) and (2) MEND can be applied after-the fact without fine-tuning the original model, meaning that anyone will be able to train and publish model editors for repositories of pre-trained models, such as HuggingFace Transformers, torch.hub.


## Strengths

- (as described above) the proposed technique has a great practical potential
- the technical contributions made in this paper are solid and significant
- the core method utilizes a clever trick (rank-1 gradients) that significantly reduces the complexity of model editing. While there are related tricks in meta-learning papers, this specific variant and application to model editing is novel
- overall, the paper is well-written and easy to follow


## Weaknesses

There are several (minor) inaccuracies in the experiments and related work sections. One of these inaccuracies may slightly mislead readers if they are unfamiliar with the related work.

__1. On the use of gradient checkpointing / rematerialization:__ the original PyTorch implementation of ENN (Sinitsin et al) heavily relies on gradient checkpointing to avoid storing too many model states in memory when computing higher-order gradients. However, authors report the memory consumption for their custom re-implementation of ENN (Figure 3, 4) without gradient checkpointing, making MEND look significantly more favourable as a result.

Using gradient checkpointing allows editing 2.8B models in 21-28GB memory instead of almost 80GB reported in Figures 3, 4. However, using ENN for 11B models still does not fit on a single A40 GPU (described in supmat), so the general conclusion holds.

To be fair, authors mention the fact that their implementation does not use gradient checkpointing (Appendix B.3), but this is only briefly mentioned in supplementary materials. Since this implementation detail improves the scalability of one of the baselines by almost 3x, i'd request that
- either authors evaluate ENN with gradient optimal checkpointing in **Figure 3 and Table 3** (for 2.7B and 2.8B models)
- or authors clearly state this as a limitation of the experiment design in the main paper

As for applying ENN to GPT-J and T5-XXL models - Sinitsin et al. propose a strategy for further reducing memory usage by editing only a subset of model layers (Section 4.3, second paragraph) -- as such, all layers prior to the first edited one can be considered constant and excluded from the unrolled meta-learning graph, further reducing memory consumption by up to an order of magnitude (if only the last transformer layer is edited). That said, this this strategy requires a careful selection of layers, so I understand why it is not evaluated.

Based on my observations, MEND still scales significantly better than ENN, especially for larger models (40B+) -- and I do understand that it is impossible to obtain enough fully trained 10B+ models in the public domain -- but i recommend including the additional comparison so as to avoid the slightest chance of misleading the reader.

__2. Minor inaccuracies in Table 1__

- "One step" - ENN is flagged as one step, but the corresponding method **does** require an interative process and original paper (Sinitsyn et al) reports 2-5 steps in most experiments.
- "Scales to 10B" - it would be preferable to refer to your hardware setup. Otherwise, all methods would comfortably "scale to 10B" on a DGX-A100 machine (640GB total GPU memory). Consider "Scales to 10B means that the method can edit a 10B model in our single-GPU setup (Appendix XYZ)"


__(lack of requirements.txt)__
I appreciate that the authors provide the source code (this affected my score positively), but I highly encourage authors to specify the exact versions of all the dependencies that they use in their experiments (i.e. pytorch, transformers, higher, etc). Without this information, it will be difficult to reproduce the experiments in the future.


### Nitpicking

This below complaints did not affect my score in either direction, and neither will addressing them.

__(Section 3.1) “the gradient … is a rank-1 matrix for each of B batch elements”__
It would be great to give an intuition of why this is the case for algebraically challenged readers. Perhaps, refer them to an appendix where you show this to be the case in a simple MLP example.

__(Ethics statement)__
There is one possible ramification of MEND that is (arguably) not covered by "treating / exacerbating undesiable behaviors": introducing model backdoors.

If somebody releases a certain model to the public domain, that somebody could use editable training to, for instance,
- tweak a face identification network to always treat a person as staff if they wear a very specific ornament -- and use it to bypass a security system
- tweak a language model to memorize some secret data and only give it away under a very specific prompt -- and then use this for steganography

As a result, MEND can actually enable a number of new outcomes, from elaborate pranks to serious opsec vulnerabilities.
To be fair, MEND can't be held solely responsible for this since it is not the only model editing method.

**Summary Of The Paper:**

This paper studies the problem of "model editing" - altering the model predictions on local examples without affecting the global behavior.
Authors tackle the challenge of editing very large models with billions of parameters, such as T5 or GPT-J, where editing is particularly important, as it is impractical to re-train such models from scratch to correct every mistake.

To address this challenge, authors propose MEND - a novel model editing method that differs from prior art in three ways:
- it does not require computing higher-order gradients (and hence, requires less compute/memory)
- it can be trained for a given model without the need to change the model parameters (and hence, can be applied to pre-existing models)
- it has better asymptotic time & memory complexity in terms of model size (and hence can be used for very large models)


**Summary Of The Review:**

I deeply appreciate the practical contributions of this paper and believe that it can have significant impact on the applicability of the large pre-trained models. That said, I do have several minor issues with the experiment design. I summarize those issues in the "Weaknesses" subsection of the main review.

---

> ### Author Response · Authors · 2021-11-10
> **Minor clarification regarding ENN gradient checkpointing**
>
> Thank you for the very detailed feedback! We will provide a complete response and paper revision soon in light of your comments. In the meantime, we wanted to clarify one point in your review. We revisited the ENN (Sinitsin et al) paper (https://openreview.net/forum?id=HJedXaEtvS) and implementation (https://github.com/xtinkt/editable) but couldn't find any references to gradient checkpointing. If possible, could you clarify the implementation of ENN with gradient checkpointing or line(s) of code you are referring to?

---

> > ### Comment · Reviewer_5ZVS · 2021-11-12
> > **I was in the wrong about the use of gradient checkpointing in the baseline**
> >
> > As it turned out, I was talking about an unofficial implementation based on this specific use of gradient checkpointing:
> > ``https://github.com/dbaranchuk/memory-efficient-maml/blob/master/torch_maml/maml.py#L61``
> >
> > the official paper and implementation indeed contain no gradient checkpointing. Therefore, while my original point about gradient checkpointing stands, it is understandable not to use it in the experimental comparison.

---

> ### Author Response · Authors · 2021-11-16
> **Response to reviewer 5ZVS**
>
> Thank you again for the in-depth review, and for your thoroughness in independently verifying MEND's scalability to a 41B parameter model. We're happy that you feel MEND has meaningful technical contributions and potential for practical impact! To address your comments:
>
> > On the use of gradient checkpointing / rematerialization...all layers prior to the first edited one can be considered constant and excluded from the unrolled meta-learning graph, further reducing memory consumption
>
> As noted in our earlier discussion, existing implementations of ENN do not implement gradient checkpointing/rematerialization to reduce memory consumption. The linked repository does enable reduced memory consumption when performing many inner loop steps, but our memory consumption curve computes the memory for ENN when using only a single inner loop step, so this optimization would not be applicable. We have clarified in the main paper (footnote 5) that the ENN memory graph uses our re-implementation, and that reducing ENN's memory consumption is possible in principle with implementation optimizations (though these optimizations are not needed for FT/KE/MEND).
>
> > As for applying ENN to GPT-J and T5-XXL models - Sinitsin et al. propose a strategy for further reducing memory usage by editing only a subset of model layers (Section 4.3, second paragraph)
>
> We have clarified in Section 5 (footnote 3) that we edit only a subset of the model parameters in our ENN implementation as well.
>
> > ENN is flagged as one step..."Scales to 10B" - it would be preferable to refer to your hardware setup
>
> We have revised Table 1 to clarify these points. Please let us know if you feel further explanation would be helpful.
>
> > I highly encourage authors to specify the exact versions of all the dependencies that they use in their experiments
>
> We completely agree- we will be releasing the fully reproducible environment & dependencies used for our experiments with the final paper!
>
> > “the gradient … is a rank-1 matrix for each of B batch elements” It would be great to give an intuition of why this is the case
>
> Thank you for this suggestion- the newly-added Appendix D provides a derivation of this identity.
>
> > There is one possible ramification of MEND that is (arguably) not covered
>
> We have added an additional sentence to our ethics statement more specifically describing this possibility.
>
> Thank you for your careful evaluation of the paper. We believe that the revisions based on your feedback have made the paper stronger. Please let us know about any remaining questions or concerns!

---

### Official Review · Reviewer_WxF5 · 2021-11-04

**Correctness:** 3
**Technical Novelty And Significance:** 2
**Empirical Novelty And Significance:** 2
**Recommendation:** 3
**Confidence:** 4

**Main Review:**

Strengths:

1. The problem of editing a pre-trained model is interesting. Since big models are deployed and getting into industrial scenarios, it is interesting to study the emerging problems they bring.

2. This paper decomposes the gradient of linear layer (size: d x d) as an outer product of (d x 1) x (1 x d), so that it operates on input with 2d dimensions rather than d^2 dimensions. This technique is reasonable, but was originally proposed by Goodfellow et al., limiting the paper's novelty.

Weaknesses:

1. The performance in small-scale model editing is on par with ENN and KE, which is not surprising. In addition, it is strange to see reported results as "<0.001".

2. One claim of this paper is the ability to edit large-scale models like T5. However, only "the MLP weight matrices in the last 2 transformer blocks" are edited, which makes the contribution an over-claim.

3. If only two blocks are edited, then how many blocks are fine-tuned in Table 3 for FT? My educated guess is that all blocks are fine-tuned. Since training with only one sentence may lead to severe over-fitting, it is natural to fine-tune only several blocks. So the baselines may not be plausible.

4. The significance of the problem is unclear. If the only purpose is to change the answer of a specific input, then a simple engineering practice can do the trick: if the representation of x is close to x_e, just output y_e. This way, the editing task can be accomplished. Did the authors try this approach?

5. I think the locality statements are somewhat over-claimed. Since a model is edited, the locality cannot be guaranteed. And I doubt if the locality can be preserved after several editing. If my understanding is correct, experiments in Table 3 edit the same model for each input. If all the datasets are used to edit the model (i.e. many edits are performed), what is the accuracy drop?

Besides, the paper says that "Uj , Vj corresponds to a low rank factorization of the weight matrix of layer j", but Figure 2 says V1/V2 are xavier initialized and U1/U2 are zero initialized. How to understand the disagreement?

**Summary Of The Paper:**

A fast model editing method is proposed in this paper to edit pre-trained models at scale. The core component is a Model Editor Networks with Gradient Decomposition (MEND). The paper claims that the MEND method has reliability, locality, generality. MEND is empirically validated on some curated datasets.

**Summary Of The Review:**

In summary, the paper studied an interesting problem but the practical significance is doubtful. The empirically results are not strong. A concern of fair comparison is raised.

---

> ### Author Response · Authors · 2021-11-16
> **Response to reviewer WxF5**
>
> Thank you for your review and interest in the model editing problem! Please let us know if these responses have addressed your questions.
>
> > This technique is reasonable, but was originally proposed by Goodfellow et al., limiting the paper's novelty.
>
> To clarify, Goodfellow et al. is a textbook showing the backprop equations for MLPs; it does not leverage the gradient factorization for learning transformations of gradients, and no other prior work has done so to our knowledge. The novelty of this paper is the fact that MEND exploits the low-rank gradient structure to learn useful transformations of the gradient.
>
> > The performance in small-scale model editing is on par with ENN and KE
>
> We have included a new comparison with ENN (which outperforms KE) with batched edits. **Even for small models**, MEND is far more effective (see Table 5 in the revised paper for complete results):
>
> | Edits | ENN ES | MEND ES | ENN DD | MEND DD |
> |-------|-----------|------------|-------|-----------|
> |5|0.94|0.97|0.007|0.005|
> |25|0.35|0.96|0.005|0.008|
> |125|0.11|0.67|0.006|0.012|
>
>
> > In addition, it is strange to see reported results as "<0.001".
>
> We use <0.001 for results that are not zero, but are smaller than the level of precision used to report our results. We felt that reporting zero in this case may be misleading.
>
> > only "the MLP weight matrices in the last 2 transformer blocks" are edited
>
> Thank you for pointing out this ambiguity; the revised abstract now states that we edit the behavior of large-scale models like T5. The revised Section 5 now also clearly states that we only edit the last few layers for all methods in all of our experiments (which was originally mentioned in Appendix B).
>
> > If only two blocks are edited, then how many blocks are fine-tuned in Table 3 for FT? My educated guess is that all blocks are fine-tuned.
>
> Our FT experiments edit the same blocks as MEND; for each dataset, the parameters that are edited are fixed to ensure fair comparison (last 3 transformer blocks for GPT/BERT, last 2 blocks in encoder and decoder for BART/T5).
>
> > If the only purpose is to change the answer of a specific input, then a simple engineering practice can do the trick
>
> We implemented this strategy; we tuned the threshold to provide similar drawdown to MEND. Caching performs well for zsRE, slightly worse for FEVER, and fails completely for Wikitext. We include the results in the following table (where epsilon indicates the threshold for determining whether the representation $x$ is close to $x_e$):
>
> | Editor | FEVER ES | FEVER DD | zsRE ES | zsRE DD | Wiki ES | Wiki DD |
> |---------|----------|-------|---------|---------|---------|---------|
> | Caching | 0.96 | <0.001 | 0.99 | <0.001 | <0.01 | 0.211 |
> | MEND | >0.99 | <0.001 | 0.98 | 0.002 | 0.86 | 0.225 |
>
> While zsRE question-answering and FEVER fact-checking generally contains short questions with one subject, one relation, and one object, Wikitext contexts are longer and contain much more diverse content, providing a much more challenging setting for a memorization-based approach, highlighting the benefits of more powerful model editors. We have noted this additional experimental comparison in the revised Section 5 and included the full results (along with ablation of the caching threshold parameter) in the revised Appendix Table 11.
>
> > I think the locality statements are somewhat over-claimed. Since a model is edited, the locality cannot be guaranteed
>
> All of the experiments report a drawdown metric that quantifies the locality of each editor, showing that MEND does provide locality that is similar or better than other editors (e.g. Table 3). The paper does not claim that MEND guarantees perfect locality (nor do other editors).
>
> > If all the datasets are used to edit the model (i.e. many edits are performed), what is the accuracy drop?
>
> Table 5 in the revised paper reports MEND's performance, including accuracy drawdown, when many edits are applied at once (using one update step to perform the entire batch of edits). For 25 simultaneous question-answering edits, MEND maintains over 95% edit accuracy, while ENN's edit accuracy drops to 35%.
>
> > Besides, the paper says that "$U_j$ , $V_j$ corresponds to a low rank factorization of the weight matrix of layer $j$", but Figure 2 says $V_1$/$V_2$ are xavier initialized and $U_1$/$U_2$ are zero initialized. How to understand the disagreement?
>
> We have clarified in Section 3.1 that $U_j$, $V_j$ correspond to a factorization of MEND's weights, not the base model's weights. $U_1$ and $V_1$ constitute the low-rank factorization of the first-layer weight matrix of the MEND editor model (not the base pre-trained model that is being edited), which learns the transformation of the base model fine-tuning gradient.
>
> In light of these new experiments and clarifications, please let us know if you have any remaining concerns about the paper! We are happy to answer any further questions you may have.

---

> > ### Comment · Reviewer_WxF5 · 2021-11-17
> > **Rebuttal is not satisfying and the model editing problem itself is in question**
> >
> > I appreciate the efforts for the rebuttal, but the rebuttal results strengthen my negative opinion on the model editing problem.
> >
> > The authors show that the simple caching trick can achieve performance on par with the proposed sophisticated MEND method on FEVER and zsRE. I think the cutting-edge direction of model editing requires a thorough re-thinking.
> >
> > As for Wiki, the problem of model editing is not not clear to me. The whole paper formulation looks like a classification problem, and so does the running example of "who is the prime minister of UK?". Caching-like tricks should work with a little attention to the specif problem.
> >
> > If a simple engineering trick can solve the model editing problem, why bother to explore such a sophisticated method without a locality guarantee?
> >
> > As for the gradient decomposition, I think it has limited novelty. As pointed out by authors, the formula has already been present in the deep learning textbook. And there is a technical report "Efficient Per-Example Gradient Computations" (arxiv 2015) with a similar formula.
> >
> > I find that I'm the only one to give a negative score, maybe because I'm not a researcher in model editing to appreciate the beauty of the proposed method. As a reviewer from another area, I paid much more attention to the practical significance.
> >
> > By the way, I have seen a surge of research interest in humongous NLP models in the last year. However, it seems these models are too large to deploy. Do you see any practical usage of these models besides pure research interest?

---

> > > ### Author Response · Authors · 2021-11-22
> > > **Significance of editing large models**
> > >
> > > Thank you for continuing to engage in a discussion during the response period. We'd like to address several points you have raised. First, we perform inference for all of the models in our experiments on a single GPU, so it's not clear what would make them particularly difficult to deploy. Further, models 10 times the size of the models in our experiments are currently being deployed:
> > >
> > > - OpenAI has deployed an API with access to a >100B parameter GPT-3 model for more than a year https://openai.com/api/, https://blog.eleuther.ai/gpt3-model-sizes/
> > >
> > > - AI21 has released a public API for building applications on top of a 178B parameter language model: https://www.ai21.com/blog/announcing-ai21-studio-and-jurassic-1
> > >
> > > - NVIDIA produces systems for deploying 500B parameter models in real-time settings: https://nvidianews.nvidia.com/news/nvidia-brings-large-language-ai-models-to-enterprises-worldwide
> > >
> > > - Microsoft is releasing an Azure AI service based on variants of 100B parameter-scale GPT-3 (including Codex), which is also already being used in their own products: https://blogs.microsoft.com/ai/new-azure-openai-service/
> > >
> > > Therefore, it is inaccurate to say that the models in our experiments are too large to deploy. In fact, they are an order of magnitude smaller than the largest models currently being deployed in industry.
> > >
> > > On the topic of our experiments, we'd also like to clarify that y_e is not a class label in the zsRE question-answering or Wikitext generation editing problems. y_e is a sequence (3 tokens long on average for zsRE and 10 tokens long for Wikitext), and predictions are generated autoregressively. Only the FEVER fact-checking problem performs classification (binary classification, specifically). We have clarified that edit targets may be sequences, not class labels, in Section 2 of our revised paper.
> > >
> > > To more clearly demonstrate the limitations of caching beyond its failure in the Wikitext setting, we have augmented the zsRE question answering problem to include some examples testing the implications of an edit. For these examples, the cached label y_e does not answer the followup question x'. For example:
> > >
> > > (x_e, y_e) = "Who is the UK PM? Boris Johnson"
> > > x', y' = "Of what state is Boris Johnson the Prime Minister? UK"
> > >
> > > The results in this regime are presented in the following table:
> > >
> > > |       | Caching | MEND |
> > > |---|---|---|
> > > |Edit success | 0.32 | **0.69** |
> > > | Accuracy drawdown | 9.5e-4 | **5.1e-4** |
> > >
> > >
> > > Here, caching succeeds only on examples that don't test implications, while MEND is able to generalize to many of the implication questions. Note that we did not re-tune any MEND hyperparameters for this setting.
> > >
> > > We also note that caching does not guarantee perfect locality, even for problems where it can perform well, unless the data are perfectly separable by the hidden states computed by the model, which is typically unlikely (and is not the case in any of our experimental settings as caching achieves less than perfect edit success with non-zero drawdown). When the data are not separable, there is a tradeoff between a small threshold (high locality, lower edit success) and high threshold (high edit success, lower locality).
> > >
> > > Finally, you are correct that we have not claimed that the mathematical fact of the gradient factorization is novel; the paper’s novelty is the application of this factorization to enabling tractable transformations of the gradient. By analogy, the backpropagation algorithm itself is a “simple” application of the chain rule, a mathematical tool known for hundreds of years. Yet, this application of the chain rule to learning representations in neural networks was clearly a novel contribution to the field of machine learning. We also note that many papers learning to transform or manipulate gradients exist [1,2,3,4,5], but none have considered leveraging the gradient factorization we use.
> > >
> > > We hope this response is helpful. In light of the increasingly common deployment of 100B+ parameter models in industry, the experimental and conceptual shortcomings of the caching approach to editing, and the comments from other reviewers regarding the significance of editing large models, we believe the model editing problem as considered in our paper is clearly motivated, challenging, and important.
> > >
> > >
> > >
> > > [1] Meta-SGD. Zhenguo Li, Fengwei Zhou, Fei Chen, Hang Li. 2017.
> > > [2] Gradient-Based Meta-Learning with Learned Layerwise Metric and Subspace. Yoonho Lee, Seungjin Choi. ICML 2018.
> > > [3] Meta-Curvature. Eunbyung Park, Junier B. Oliva. NeurIPS 2019.
> > > [4] Meta-Learning with Warped Gradient Descent. Sebastian Flennerhag, Andrei A. Rusu, Razvan Pascanu, Francesco Visin, Hujun Yin, Raia Hadsell. ICLR 2020.
> > > [5] On Modulating the Gradient for Meta-Learning. Simon, Christian and Koniusz, Piotr and Nock, Richard and Harandi, Mehrtash. ECCV 2020.

---

> > > > ### Comment · Reviewer_WxF5 · 2021-11-23
> > > > **Continuing the discussion**
> > > >
> > > > Thanks for the detailed response.
> > > >
> > > > I have read those blogs for deploying large models. In my personal opinion, it seems they are just bragging about the capacity to train and deploy large models, without discussing the necessity of large models. My concern is that, how much improvement does large models bring, and what is the cost of the improvement (in flops or in consumed power energy)?
> > > >
> > > > For the caching setting, the example you propose is not reasonable. In the example (x_e, y_e) = "Who is the UK PM? Boris Johnson" x', y' = "Of what state is Boris Johnson the Prime Minister? UK", **x', y' are not the neighborhood of (x_e, y_e)** I think. Maybe it would be better to formally define what is a neighborhood. Mathematically speaking, neighborhood means a small area around a point, that's why I think caching should be good enough: transform x_e into a vector, and inputs with similar representations are considered in the neighborhood.
> > > >
> > > > For the gradient factorization, would it be more precise to say that the gradient is rank-B (B is the batch-size)? If you say gradient is rank-1, then do you mean batch-size B is set to be 1 for all the experiments?

---

> > > > > ### Author Response · Authors · 2021-11-24
> > > > > **Re: continued discussion**
> > > > >
> > > > > Regarding scaling, there is significant evidence that these models provide substantial improvements while maintaining manageable inference-time costs. For example, the [GPT-3 paper](https://arxiv.org/abs/2005.14165) has multiple experiments that show quantitative evidence of large improvements from scaling such as in Figure 3.3. Regarding inference costs, the API pricing from [OpenAI](https://openai.com/api/pricing/) and [Cohere.ai](https://cohere.ai/pricing) indicates that the 175B-equivalent model is 6 cents per 1000 tokens, which is at all not impractical or unreasonable. Regardless of one's opinion, we do not think that the study of large models should be grounds for rejecting a paper.
> > > > >
> > > > > When we use the term neighborhood, we consider the same definition as you provide (small area around a point in some vector space), but notably the vector space is not known — it must be inferred from the training dataset of edits. The example we provide *does* fall under this definition of neighborhood, i.e. there exists a vector space where these two datapoints are very close, and a neural network can transform semantic notions of neighborhoods into precise mathematical ones. Importantly, we do not expect caching to work in general when the vector space is unknown because the representation from the trained model may not capture the desired notion of neighborhood. The wikitext problem and this new problem are settings where the trained model does not produce a good vector space for the kind of edits in the evaluation set, which can explain the poor performance of caching. MEND on the other hand can perform well on both problems by using the training dataset of edits to implicitly infer the desired notion of neighborhood.
> > > > >
> > > > > When the batch size is greater than 1, we apply the model editor to the rank-1 gradients of each example independently, and then sum the resulting updates. This is described in footnote 2 on page 4 and in Section 5.3.
> > > > >
> > > > > Thank you for your continued discussion, and please let us know if you have any additional feedback on the paper.

---

> > > > > > ### Comment · Reviewer_WxF5 · 2021-11-24
> > > > > > **continued discussion**
> > > > > >
> > > > > > Regarding scaling, it is a side discussion and my score is not based on the significance of scaling large models.
> > > > > >
> > > > > > Regarding the batch-size and rank of gradient, it is a technique detail and do not affect my score.
> > > > > >
> > > > > > The point I concern most is that the performance of a sophisticated MEND method is almost the same in zsRE and FEVER as a training-free & learning-free caching method.
> > > > > >
> > > > > > You mention that the neighborhood is defined by the dataset, which I disagree. I think the neighborhood should be inherent. A proper dataset should respect the inherent neighborhood. The dataset you later propose is not a valid "dataset" for model editing. For example,  (x_e, y_e) = "Who is the UK PM? Boris Johnson" x', y' = "Of what state is Boris Johnson the Prime Minister? UK", x', y' are not the neighborhood of (x_e, y_e).
> > > > > >
> > > > > > If the inherent neighborhood cannot be well defined, I think model editing is not a well-defined problem.

---

### Official Review · Reviewer_tLcY · 2021-11-09

**Correctness:** 3
**Technical Novelty And Significance:** 2
**Empirical Novelty And Significance:** 2
**Recommendation:** 6
**Confidence:** 3

**Main Review:**

The main contribution of the paper is in leveraging the fact that gradients with respect to the fully-connected layers in neural networks are rank-1, enabling a parameter-efficient architecture that represents this gradient transform. This leverage results in an efficient lightweight model editor networks to produce edits to a pre-trained model’s weights when provided with the standard fine-tuning gradient of a given correction as input. The other strnegth of the paper is in justifying the architecture of the proposed MEND model through ablation studies, and showing its advantage over the existing methods through multiple numerical experiments.

The paper can be improved by better writing, particularly of the methodology section. In its current form, it is difficult to understand the exact loss function of the model editor networks. Further, there seems to be a lot of notations in the paper, some of which are redundant.

**Summary Of The Paper:**

While large pre-trained nlp models have achieved great performance on a variety of downstream tasks, the largest of these models still make errors. This paper deals with the problem of enabling both developers and end users of such models to correct inaccurate outputs while leaving the model otherwise intact. If presented with only a single problematic input and new desired output, fine-tuning approaches tend to overfit. To enable easy post-hoc editing at scale, this paper proposes Model Editor Networks with Gradient Decomposition (MEND), a collection of small auxiliary editing networks that use a single desired input-output pair to make fast, local edits to a pre-trained model.

The approach trains lightweight model editor networks to produce edits to a pre-trained model’s weights when provided with the standard fine-tuning gradient of a given correction as input, leveraging the gradient as an information-rich starting point for editing.

MEND leverages the fact that gradients with respect to the fully-connected layers in neural networks are rank-1, enabling a parameter-efficient architecture that represents this gradient transform.

It provides experiments with T5, GPT, BERT, and BART models showing that MEND is the only approach to model editing that produces effective edits for models with tens of millions to over 10 billion parameters.

**Summary Of The Review:**

The paper provides an efficient approach for correcting inaccurate outputs of large nlp models while leaving the model otherwise intact. They show the advantage of their proposed approach over the existing methods through a number of numerical experiments.

---

> ### Author Response · Authors · 2021-11-16
> **Response to reviewer tLcY**
>
> Thank you for your review!
>
> We're glad you appreciated our numerical experiments and ablation studies. Regarding loss functions, the original paper describes MEND's training losses in Equations 4a and 4b in Section 3.1, as well as in lines 5, 6, and 7 of Algorithm 1. We have further highlighted the loss functions presented in Equations 4a and 4b in the revised paper to make them easier to identify- we hope this addresses your concern. If there is specific notation that is difficult to parse, please let us know, and we'll be happy to revise it!

---

### Author Response · Authors · 2021-11-16
**Summary of revisions**

We thank the reviewers for their thoughtful and thorough comments.

All reviewers appreciated the problem setting of editing large models, and 5ZVS and vuAW found our proposed methodology "solid and significant" and "elegant and effective," respectively. Reviewer WxF5 requested an additional baseline as well as clarifications of some experimental protocols. We are excited that reviewer 5ZVS was able to run MEND on a 41B parameter transformer, almost 4x the size of our largest experimental model.

Based on the feedback from all reviewers, we have performed new experiments and revised some writing to make the claims and experimental protocols more precise, which we feel has further strengthened the paper and highlighted the practical utility of MEND. Our changes to the paper text can be found in the revised pdf in $\textcolor{blue}{\text{blue}}$. To summarize, our main changes include:

- Comparison with ENN with more than one edit. Even for a small model, MEND provides a much higher edit success rate than ENN when more than one edit is needed, with slightly worse drawdown. See Table 5 in the revised paper for the complete comparison.
- Additional ablation studies showing that editing attention layers underperforms editing weight matrices in Appendix E.
- Additional qualitative examples in Appendix F, showing additional success and failure cases of MEND
- An additional baseline suggested by WxF5, which involves caching the average hidden representation for an edit example sequence, and outputting the edit label for new inputs with similar input representations. We find that this heuristic performs slightly better than MEND for QA slightly worse for fact-checking, and fails completely for generative modeling. We note these findings in Section 5 of the revised paper and include the full results in Appendix G.
- Clarification in the beginning of Section 5 that in our experiments, all methods edit the same subset of the model parameters. We have also clarified which parameters are edited for each model.
- Clarification in the abstract that we edit the *behavior* of large models

We hope that these additional textual clarifications and experimental results have addressed the reviewers' concerns. Please let us know if there are any lingering concerns or questions.

---

### Comment · Reviewer_WxF5 · 2021-12-02
**Clarify my opinion**

I'm not opposed to using large models. The question of using large models is only a side question I want to discuss with the authors.

My review score is based on the fact that the proposed sophisticated method would not be as practical as a simple engineering trick (i.e. caching).

To further clarify my standpoint, I would not mind if this paper is accepted. However, I require a detailed discussion about the practical significance of model editing in a future revision, given the effectiveness and training-free characteristic of caching.

The model editing problem is interesting, and if someone really wants to solve such a problem, this paper should tell him that in some scenario caching can be a choice besides time-consuming training-based method.

---

> ### Comment · Reviewer_5ZVS · 2021-12-02
> **Strongly agree**
>
> I strongly support WxF5's request that the caching trick be introduced and discussed in the final revision. While this trick has some fundamental limitations (e.g. the transformer would substitute a given word regardless of any context that is not captured in final layer's embeddings for that word), it is unclear (1) how often do these limitations actually affect model performance and (2) if the existing evaluation setup is sensitive to them.
>
> Even if the caching trick ends up having worse metrics, it would indeed be preferable for many practical applications due to its simplicity, e.g. if there are not enough development resources to justify introducing model editors (e.g. if the model is not the main business value). In turn, MEND would be easier to justify for large LM applications where performance is more important.
>
> To better explain my perspective, below I describe one practical application where the extra complexity of model editing is (arguably) justifiable. Disclaimer: I am not directly involved in this application, but I closely communicate with at least 2 developer teams working on this system in at least 2 companies. (Previous sentence is formulated weirdly to preserve anonymity)
>
> The purpose of this application is to let a non-ML-expert quickly build and deploy NLP solutions to their problem without the need for labeling datasets and/or heavy engineering. To the best of my knowledge, all pipelines for solved with the following pipeline:
>
> 1. User (developer) creates a very small dataset (10s to 100s of examples) for their problem. This can usually be done in a day.
> 2. User feeds this dataset to an automated prompt tuning system that adapts a (large) language model to solve their problem [1, 2];
> 3. Once prompt tuning model achieves satisfactory quality, it is automatically deployed as an API or distilled into a smaller model for high-throughput tasks [3];
>
> **Q1: What is the purpose of this system? A:** It allows any practitioner (e.g. an analyst or a start-up creator w/o experience in training or deploying transformers) to develop a working ML solution in a day and quickly test their ML scenario in practice. In contrast, fine-tuning and deploying custom models normally takes weeks and requires DL expertise.
>
> **Q2: Does this application need large models? A:** Arguably, yes. Prompt-tuning was shown to be particularly effective for large models [2]. In turn, it is usually not a problem that the model will be slow to inference because most prototypes do not require high throughput. However, if performance is critical, large models also allow for efficient distillation through data [3] that can be used to automatically create (and deploy) more efficient models.
>
> **Q3: Why does this benefit from model editing? A:** Editing allows one to fix specific undesirable behaviors, which is important in prototyping. This is on top of normal use cases such as filtering illicit content from the base model.
>
> **Q4: Is it justified to use (complex) model editing rather than simple engineering tricks? A:** Arguably, it is. A given model editing solution can be developed once and reused for all subsequent prototypes. The added complexity of system-level model editing is (arguably) not a significant concern because such system already requires several similar sub-modules (e.g. a prompt tuning interface).
>
> [1] All nlp tasks are generation tasks: A general pretraining framework. arXiv preprint arXiv:2103.10360.
>
> [2] The Power of Scale for Parameter-Efficient Prompt Tuning https://arxiv.org/abs/2104.08691
>
> [3] Symbolic Knowledge Distillation: from General Language Models to Commonsense Models https://arxiv.org/pdf/2110.07178.pdf

---

### Author Response · Authors · 2021-12-03
**General comment in response to followup discussions**

We really appreciate the reviewers' continued discussion of the paper, which we think has been productive. We agree that the caching baseline as suggested by WxF5 provides useful context for the evaluation of MEND and other approaches to learning to edit. We will include the performance of this baseline in the main text of the final draft, as well as a discussion of when caching versus learning to edit approaches are the most appropriate.

Thank you again for the discussion. We believe that the paper is stronger as a result!

---

### Decision · Program_Chairs · 2022-01-20

**Decision:**

Accept (Poster)

**Comment:**

The paper provides a method to edit trained models, meaning fix mistakes in a local way so as to not ruin generalizability. The techniques provided in the paper allow for an efficient way that makes this task possible for very large models.
There is an overall concensus that the problem of model editing in general is an important one and that solutions such as naive finetuning are not applicable for various reasons. In addition, the reviewers are convinced that given the need for an ML-based approach for large models, this technique is superior to previous work, and mostly appreciate the novelties of the paper.

A major concern raised regards possible simpler baselines. There is a potential baseline of implementing an “engineering trick” that will simply memorize the data points where the original model was mistaken, either in their original form or as embeddings, and during inference will override its output. I tend to agree that a comparison with such a baseline would improve the paper. This being said, the discussion highlighted that this baseline has several flaws that make it clear that it cannot completely replace the method proposed here. A naive implementation of it will be “too local” and would not handle simple rephrasing of sentences. An implementation operating on the embedding space will be possible only in a subset of tasks.

To conclude, although the paper has room for some improvement (that might actually be possible towards the camera-ready version), I believe that even without it the paper is in a good enough state to be published. It tackles an important problem and could lead to further advancements.